# CONTINUAL UNSUPERVISED DISENTANGLING OF SELF-ORGANIZING REPRESENTATIONS

**Zhiyuan Li**[1], **Xiajun Jiang**[1], **Ryan Missel**[1], **Prashnna Kumar Gyawali**[2],
**Nilesh Kumar**[1], **Linwei Wang**[1]
Rochester Institute of Technology[1], Stanford University[2]
`{zl7904,xj7056,rxm7244,nk4856,Linwei.Wang}@rit.edu`,
`pgyawali@stanford.edu`

## ABSTRACT

Limited progress has been made in continual unsupervised learning of representations, especially in reusing, expanding, and continually disentangling learned semantic factors across data environments. We argue that this is because existing approaches treat continually-arrived data independently, without considering how they are related based on their underlying semantic factors. We address this by a new generative model describing a topologically-connected mixture of spike-and-slab distributions in the latent space, learned end-to-end in a continual fashion via principled variational inference. The learned mixture automatically discovers the active semantic factors underlying each data environment, and to accordingly accumulate their relational structure. This distilled knowledge can further be used for generative replay and guiding continual disentangling of sequentially-arrived semantic factors. We tested the presented method on a split version of 3DShapes to provide the quantitative disentanglement evaluation of continually learned representations, and further demonstrated its ability to continually disentangle new representations and improve shared downstream tasks in benchmark datasets.

## 1 INTRODUCTION

The progress in continual learning has been mostly made for supervised discriminative learning, whereas continual unsupervised representation learning remains relatively under-explored (Ramapuram et al., 2020; Achille et al., 2018; Rao et al., 2019). The few existing works have primarily focused on battling *catastrophic forgetting* in the generative performance of a model: for instance, a common approach known as *generative-replay* synthesizes *past* samples using a snapshot of the generative model trained from past data, and then continually trains the model to generate both new data and synthesized past samples (Achille et al., 2018; Rao et al., 2019; Ramapuram et al., 2020).

There is however another important yet under-explored question in continual unsupervised representation learning: how to *reuse, expand, and continually disentangle* latent semantic factors across different data environments? These are inherent in the human learning process: while learning from new data (*e.g.*, learning cars after bicycles), we are naturally able to reuse shared semantic factors without re-learning (*e.g.*, wheels), expand and disentangle new semantic factors (*e.g.*, the shape of cars), while accumulating knowledge about the relationship among data environments based on these semantic factors (*e.g.*, bicycles and cars both have wheels but are different in shapes). Disentangled representation learning, as a long-standing research topic, has demonstrated various benefits in generative modeling and downstream tasks (Higgins et al., 2017; Kumar et al., 2017; Kim & Mnih, 2018; Liu et al., 2021; Rhodes & Lee, 2021; Horan et al., 2021). With increasing recent interests in unsupervised representation learning in a continual learning setting (Rao et al., 2019; Madaan et al., 2021), it is important to investigate the challenges and solutions to achieve disentanglement of sequentially-arrived semantic factors in streaming data.

*Reusing* latent dimensions for learned semantic factors has mainly been attempted by a teacher-student like approach where the student model is taught to infer and generate similarly to a snapshot of the past models (teacher) on *replayed* data (Achille et al., 2018; Ramapuram et al., 2020). In Achille et al. (2018), this is further facilitated by explicitly masking out latent dimensions that are not actively used in a data environment. Such masks however have to be heuristically defined *before* training on

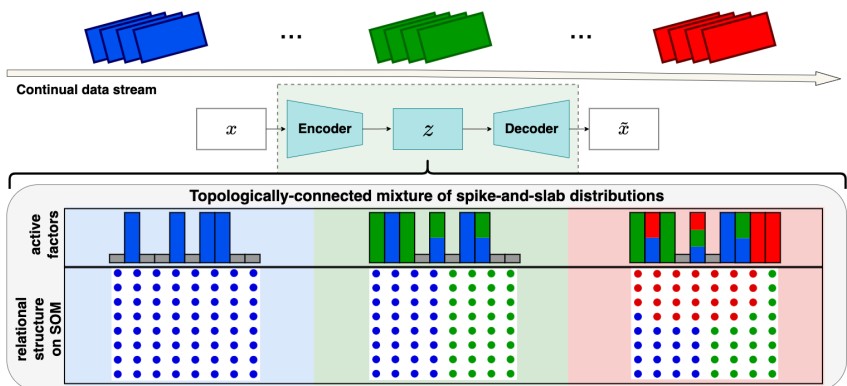

Figure 1: Through a self-organizing spike-and-slab mixture, CUDOS continually distills knowledge about the relational structure of data environments with their shared and distinct semantic factors.

the new environment. How to automatically discover latent dimensions explaining *active* semantic factors underlying each data environment, in a continual fashion, remains an open question.

*Expanding* learned semantic factors, in part, results naturally from continually optimizing a generative model to newly arrived data (Achille et al., 2018; Ramapuram et al., 2020), or even progressively increasing the model's latent capacity on the same data (Burgess et al., 2017; Li et al., 2020). These alone however would not uncover the relational structure among data environments in the latent space. In Rao et al. (2019), a mixture of Gaussians has been used and continually expanded such that newly arrived data are clustered to an existing or new mixture component. While this captures the expanding data distributions, it does not consider the reuse of semantic factors among data clusters.

*Continually disentangling* semantic factors, until now, is limited to the native disentangling ability inherent in VAE, or promoting the reusing of shared semantic factors (Achille et al., 2018; Ramapuram et al., 2020). While the common strategy of generative replay teaches a model what latent dimensions to use for shared semantic factors on the replayed data (Achille et al., 2018; Ramapuram et al., 2020), no such guidance is available on new data. As a result, as we will show, none of the existing approaches can prevent newly-learned semantic factors to be entangled with re-used ones.

In this work, we show that the above limitations boil down to a fundamental bottleneck in existing continual unsupervised learning of representations: that the learner is asked to treat continually-arrived data independently, *without knowing how they are related based on the underlying semantic factors.* To overcome this, we argue that the model needs to learn two critical knowledge: latent dimensions explaining *active* semantic factors underlying each data environment, and the relationship among the latter based on the former. We present Continual Unsupervised Disentangling of self-Organizing representations (CUDOS) that is able to accumulate the relational structure of continually-arrived data based on their underlying *active* semantic factors, and exploiting this knowledge to guide disentangling of sequentially-arrived semantic factors. As illustrated in Fig. 1, to accumulate the relational structure of the data, we model the latent representations with a topologically-connected mixture of distributions via Bayesian self-organizing maps (SOM) (Kohonen, 1990; Yin & Allinson, 2001). To automatically discover *active* semantic factors underlying each data environment, we model each component of the SOM mixture with a spike-and-slab distribution (Titsias & Lazaro-Gredilla, 2011; Tonolini et al., 2020), such that the sparse spike variable identifies latent dimensions explaining active semantic factors. This results in a generative model with a self-organizing mixture of slab-and-spike distributions, where the distilled knowledge – the relational structure of data environments and their associated active semantic factors – supports 1) mixture-based generative replay and 2) continual disentangling of sequentially-arrived semantic factors.

We evaluated CUDOS on both benchmark datasets for continual representation learning, and a split version of 3DShapes (Burgess & Kim, 2018) designed for quantitative evaluation of disentangling sequentially-arrived semantic factors. In comparison to existing works, we showed that CUDOS not only addressed catastrophic forgetting, but also improved – both quantitatively and qualitatively – continual disentanglement of latent semantic factors and thereby downstream discriminative tasks.

## 2 RELATED WORKS

**Deep Learning with SOM:**The use of SOM has been explored within deep learning of image classification (Liu et al., 2015), image clustering (Manduchi et al., 2019; Forest et al., 2000), and time-series prediction (Jin et al., 2015; Fortuin et al., 2019). None of these works, however, considered continual learning of representations. In the context of continual learning, SOM was used to first learn the relationship among all discriminative tasks, and then used to mask the neurons of a fully-connected layer for each task (Bashivan et al., 2018). A dual-memory self-organizing architecture was also presented for learning object instances and categories in life-long object recognition (Parisi et al., 2018). While both works shared our motivation to leverage the ability of SOM to maintain and expand a memory of data distributions across environments, neither considered continual unsupervised representation learning, or utilized SOM as a topologically-connected Bayesian mixture model.

**Inferring Active Semantic Factors in VAE:** Learning meaningful representations is a vital task for VAE, where large latent space often leads to latent dimensions that carry little information (Chen et al., 2017). Sparse coding and discrete latent space have proved to be elegant solutions (Oord et al., 2017; Tonolini et al., 2020). Non-parametric discrete densities have been explored for stochastic latent activation to improve disentanglement (Gyawali et al., 2019). A discrete latent space based on vector quantization was shown to solve posterior collapse (Oord et al., 2017), whereas sparsity was directly modeled in a continuous latent space using spike-and-slab priors (Tonolini et al., 2020). None of these concepts has been extended to continual unsupervised learning of representations.

**Continual Unsupervised Representation Learning:** Most existing works in continual unsupervised representation learning relied on enhanced generative replay to combat catastrophic forgetting of generation using learned semantic factors (Achille et al., 2018; Ramapuram et al., 2020; Ye & Bors, 2020). The two most related works are those presented in Achille et al. (2018) and Rao et al. (2019). In Achille et al. (2018), to reuse latent dimensions explaining semantic factors shared with past data, a mask for active latent dimensions specific to each data environment was used. This mask was heuristically determined *before* training on the new data, which will affect the learning and sharing of latent dimensions on the new data. Furthermore, while this strategy protects latent dimensions specific to past data environments (not shared and thus turned off), it does not prevent new semantic factors from being entangled with the shared dimensions. CUDOS presents fundamental solutions to these problems by 1) principled variational inference of *active* latent dimensions leveraging slab-and-spike priors, and 2) guiding continual disentangling of sequentially-arrived semantic factors by exploiting the relational structure of data.

In Rao et al. (2019), a mixture of Gaussians was continually expanded as new data arrive. This shares our intuition that continual learning of representations can benefit from accumulating an evolving summary of data in the latent space. How the different data clusters are related in terms of shared and distinct semantic factors, however, was not exploited in Rao et al. (2019). In contrast, CUDOS exploits the relationship among data clusters based on their underlying active semantic factors, and uses that to facilitate continual reuse and disentangling of sequentially-arrived semantic factors.

## 3 METHODS

We first establish the foundation for CUDOS: a VAE with a self-organizing mixture of spike-and-slab priors to learn the relational structure of data based on their *active* semantic factors (Section 3.1). We then describe how to use this data summary for generative replay (Section 3.2.2), and use the relation between new and past data to improve continual disentangling of representations (Section 3.2.3).

### 3.1 LEARNING THE RELATIONAL STRUCTURE OF DATA VIA ACTIVE SEMANTIC FACTORS

Fig. 1 outlines the foundation model underlying CUDOS. Given data $\mathbf{x}$, we are interested in learning representations of meaningful semantic factors within a latent vector $\mathbf{z} \in \mathbb{R}^J$. While doing so we encourage sparse coding to discover latent dimensions explaining *active* semantic factors in VAE (Tonolini et al., 2020), while learning the relational structure of data based on these semantic factors.

**Generative Model:** To accumulate the relational structure of data, we model the latent space by a mixture of distributions using Bayesian-SOM (Yin & Allinson, 2001) with $K$ nodes. Additionally, to discover the active semantic factors for each data environment, we model each mixture component as

a spike-and-slab distribution that encourages sparsity in the latent dimensions (Tonolini et al., 2020). This gives rise to the following generative process:

$$w \sim \text{Cat}(\boldsymbol{\pi}), \quad p_{\psi_k}(\mathbf{z}|w_k = 1) = \prod_{j=1}^{J} [\alpha_k^j \mathcal{N}(z^j; \mu_k^j, (\sigma_k^j)^2) + (1 - \alpha_k^j)\delta(z^j)],$$
(1)
$$p_\psi(\mathbf{z}) = \sum_{k=1}^{K} p(\mathbf{z}|w_k)p(w_k), \quad \mathbf{x} \sim p_\theta(\mathbf{x}|\mathbf{z}), \quad p_{\theta,\psi}(\mathbf{x}, \mathbf{z}, w) = p_\theta(\mathbf{x}|\mathbf{z})p_\psi(\mathbf{z}|w)p(w).$$

where $*^j$ denotes the $j$-th element of $*$, and $\delta(*)$ Dirac delta function centered at zero. The mixing prior $p(w)$ is parameterized by $\boldsymbol{\pi}$, $\pi_i \geq 0$ and $\sum \pi_i = 1$. The latent variable $\mathbf{z}$ from the $k$-th mixture component is parametrized by $\psi_k = \{\boldsymbol{\mu}_k, \boldsymbol{\sigma}_k^2, \boldsymbol{\alpha}_k, \pi_k\}$, where parameter $\boldsymbol{\alpha}_k$ introduces sparsity to mask out inactive dimensions. Data $\mathbf{x}$ are generated from $\mathbf{z}$ via a neural network parametrized by $\theta$.

**Inference Model:** We define variational approximations of the posterior density $p(\mathbf{z}, w|\mathbf{x})$ as:

$$q(\mathbf{z}, w|\mathbf{x}) = q_\phi(\mathbf{z}|\mathbf{x})p_\psi(w|\mathbf{z}), \quad q_\phi(\mathbf{z}|\mathbf{x}) = \prod_{j=1}^{J} [\tilde{\alpha}^j \mathcal{N}(z^j; \tilde{\mu}^j, (\tilde{\sigma}^j)^2) + (1 - \tilde{\alpha}^j)\delta(z^j)],$$
(2)
$$p_\psi(w_k = 1|\mathbf{z}) = \frac{p(\mathbf{z}|w_k, \psi_k)p(w_k = 1)}{\sum_{k'=1}^{K} p(\mathbf{z}|w_{k'}, \psi_{k'})p(w_{k'} = 1)}.$$

where $p_\psi(w_k = 1|\mathbf{z})$ is the posterior probability of the $k$-th mixture component. For parameters $\tilde{\boldsymbol{\mu}}$ and $\log \tilde{\boldsymbol{\sigma}}^2$ of the spike-and-slab distribution, their inference is amortized as the output of an encoding network parameterized by $\phi$. For parameter $\tilde{\boldsymbol{\alpha}}$, considering that similar data should share latent dimensions for common semantic factors, we infer it at a set level as inspired by the *neural statistician* (Edwards & Storkey, 2017; Hewitt et al., 2018). We discuss how to determine the set in Section 3.2.1.

**Variational Inference:** The parameters $\psi$, $\theta$, and $\phi$ are optimized by the ELBO loss as:

$$\log p(\mathbf{x}) \geq \mathcal{L}_{\text{ELBO}} = \mathbb{E}_{q_\phi(\mathbf{z}, w|\mathbf{x})}[\log \frac{p_{\theta,\psi}(\mathbf{z}, \mathbf{x}, w)}{q_\phi(\mathbf{z}, w|\mathbf{x})}] = \mathbb{E}_{q_\phi(\mathbf{z}, w|\mathbf{x})}[\log \frac{p_\theta(\mathbf{x}|\mathbf{z})p_\psi(\mathbf{z}|w)p(w)}{q_\phi(\mathbf{z}|\mathbf{x})p_\psi(w|\mathbf{z})}]$$
$$= \mathbb{E}_{q_\phi(\mathbf{z}|\mathbf{x})}[\log p_\theta(\mathbf{x}|\mathbf{z})] - \mathbb{E}_{p_\psi(w|\mathbf{z})}[D_{KL}(q_\phi(\mathbf{z}|\mathbf{x})||p_\psi(\mathbf{z}|w))] - \mathbb{E}_{q_\phi(\mathbf{z}|\mathbf{x})}[D_{KL}(p_\psi(w|\mathbf{z})||p(w))].$$
(3)

The first reconstruction term is similar to that in the vanilla VAE. The second term measures the KL-divergence between $q_\phi(\mathbf{z}|\mathbf{x})$ and its conditional prior, measured over the posterior distribution of the SOM mixture $p_\psi(w|\mathbf{z})$. Specifically, we estimate the expectation over $p_\psi(w|\mathbf{z})$ as:

$$\mathbb{E}_{p_\psi(w|\mathbf{z})}[D_{KL}(q_\phi(\mathbf{z}|\mathbf{x})||p_\psi(\mathbf{z}|w))] = \sum_{k=1}^{K} p_\psi(w_k = 1|\mathbf{z})D_{KL}[q_\phi(\mathbf{z}|\mathbf{x})||p_\psi(\mathbf{z}|w_k = 1)], \quad (4)$$

where $p_\psi(w_k|\mathbf{z})$ can be computed in a batch during forward propagation, and $D_{KL}[q_\phi(\mathbf{z}|\mathbf{x})||p_\psi(\mathbf{z}|w_k = 1)]$ can be derived following (Tonolini et al., 2020) as:

$$\sum_{j}^{J} [\tilde{\alpha}^j (\log \frac{\sigma_k^j}{\tilde{\sigma}^j} + \frac{(\tilde{\mu}^j - \mu_k^j)^2 + (\tilde{\sigma}^j)^2}{2(\sigma_k^j)^2} - \frac{1}{2}) + (1 - \tilde{\alpha}^j)\log \frac{1 - \tilde{\alpha}^j}{1 - \alpha_k^j} + \tilde{\alpha}^j \log \frac{\tilde{\alpha}^j}{\alpha_k^j}]$$

This KL loss encourages $q_\phi(\mathbf{z}|\mathbf{x})$ to follow a mixture density with the mixing probability determined by the posterior probability of each component given $\mathbf{z}$. Note that the latter automatically considers sharing of semantic factors between the inferred $\mathbf{z}$ and each mixture component $k$ (via spike variable $\tilde{\boldsymbol{\alpha}}$ and $\boldsymbol{\alpha}_k$). The third term in Eqn. (3) measures the KL-divergence between the posterior density of $w$ and its prior (set to be uniform in this work). The expectation is estimated by Monte Carlo samples.

**Iterative Optimization:** We maximize the ELBO loss as defined in Eqn. (3) by iterative optimization. In each iteration, we first fix $\psi$ of the SOM mixture and maximize Eqn. (3) with respect to the VAE's parameters $\theta$ and $\phi$ by stochastic gradient descent with reparameterization trick (Kingma & Welling, 2014; Tonolini et al., 2020): at the first iteration, the SOM-mixture is initialized as a uniform mixture of $\psi = \{\mathbf{0}, I, \mathbf{0.2}\}$ and the optimization becomes a standard ELBO with spike-and-slab priors.

Figure 2: (a) Combating catastrophic forgetting by generative replays from SOM-mixture, and (b) Continually disentangling by using the relation between new and past data based on their underlying shared semantic dimensions.

With the updated $\theta$ and $\phi$, we then maximize Eqn. (3) with respect to the SOM-mixture parameter $\psi$ which, as derived in Appendix-I, amounts to maximizing the expectation of the log-likelihood of $p_\psi(\mathbf{z})$ over the variational posterior distribution of $q_\phi(\mathbf{z}|\mathbf{x})$: $\psi^* = \operatorname{argmax}_\psi \mathbb{E}_{q_\phi(\mathbf{z}|\mathbf{x})}[\log p_\psi(\mathbf{z})]$ We follow the theory in Gepperth & Pfülb (2021) to optimize $\psi$ using stochastic gradient descent.

## 3.2 CONTINUAL LEARNING WITH CUDOS

We now consider a setting of continual unsupervised learning where the label of the underlying data environments is unknown. Following existing approaches (Achille et al., 2018; Rao et al., 2019; Ramapuram et al., 2020), we maintain a snapshot of the model parameters $[\psi_{\text{old}}, \theta_{\text{old}}, \phi_{\text{old}}]$ every $\tau$ training steps. These model snapshots are used to guide (1) synthesizing replayed samples to teach the model to perform consistently on past data (Section 3.2.2), and (2) continually disentangling sequentially-arrived semantic factors using the relationship between past and new data (3.2.3).

### 3.2.1 INFERRING $\alpha$ FOR STREAMING DATA

As mentioned in Section 3.1, we choose to infer $\tilde{\alpha}$ at a set level to leverage shared information underpinning a set of data. If the boundary between data environments is known, $\tilde{\alpha}$ can be shared for all data within the same data environment. Alternatively, if the boundary of data environments is not known, we assume $\tilde{\alpha}$ to be shared within each mini-batch $\mathbf{x}_{\text{batch}}$. Specifically, we maintain $\phi_\alpha = \{\tilde{\alpha}_1, \tilde{\alpha}_2, \cdots, \tilde{\alpha}_d\}$ for $d$ number of sets with distinct underlying semantic factors. Given a new batch of $\mathbf{x}_{\text{batch}}$, we first determine if it can be described by an existing $\tilde{\alpha}$, or a new $\tilde{\alpha}_{d+1}$ has to be allocated if the existing $\phi_\alpha$ fails to describe $\mathbf{x}_{\text{batch}}$ well:

$$\tilde{\alpha}_{\text{new}} = \begin{cases} \hat{\alpha}, & \text{if } \mathbb{E}_{p(\mathbf{z}|\tilde{\mu}, \tilde{\sigma}^2, \hat{\alpha})}[\mathcal{L}_r] \leq T_\alpha, \\ \tilde{\alpha}_{d+1}, & \text{otherwise,} \end{cases} \tag{5}$$

where $\hat{\alpha} = \operatorname{argmin}_{\tilde{\alpha} \in \phi_\alpha} \mathbb{E}_{p(\mathbf{z}|\tilde{\mu}, \tilde{\sigma}^2, \tilde{\alpha})}[\mathcal{L}_r]$, $T_\alpha$ is a threshold for the reconstruction error $\mathcal{L}_r$ of $\mathbf{x}_{\text{batch}}$ averaging over pixels. Note that Equation (5) only determines how a data batch $\mathbf{x}_{\text{batch}}$ is associated with a variable $\tilde{\alpha}$. The values of these variables are optimized during variational inference.

### 3.2.2 GENERATIVE REPLAY WITH CUDOS

To combat forgetting, we synthesize data samples following the generation process as defined in Eqn. (1), using the snapshot of the SOM mixture with parameters $\psi_{\text{old}}$. As illustrated in Fig. 2(a), on the synthesized samples $\tilde{\mathbf{x}}_{\text{old}}$ and their corresponding latent samples $\mathbf{z}_{\text{p}}$, we encourage the model to be consistent with its past snapshot (Ramapuram et al., 2020; Achille et al., 2018; Lezama, 2019):

$$\mathcal{L}_{\text{old}} = \mathcal{L}_c[S(p_\theta(\mathbf{x}|\mathbf{z}_{\text{p}})), \tilde{\mathbf{x}}_{\text{old}}] + \mathcal{L}_c[S(q_\phi(\mathbf{z}|\tilde{\mathbf{x}}_{\text{old}})), \mathbf{z}_{\text{p}}], \tag{6}$$

where $\mathcal{L}_c$ is mean squared error and $S(\cdot)$ is a sampling process. This strategy combines the two key existing concepts in *replay mechanism*: as in generative replay (Shin et al., 2017; Ramapuram et al., 2020), data are synthesized quickly and readily with limited burden on storage; as in core-set methods (Nguyen et al., 2018; Borsos et al., 2020), the SOM-mixture provides a representative summary of data ensuring that more important mixture components are more frequently re-used in future training.

### 3.2.3 GUIDING CONTINUAL DISENTANGLEMENT

Generative replay lacks mechanisms to teach the model how to disentangle newly-arrived semantic factors from latent dimensions already used for previously-learned semantic factors. CUDOS provides

a unique opportunity to address this issue by its ability to describe the relation between new and past data. For a new data $\mathbf{x}$, its shared latent dimensions with SOM summary of past data (parameterized by $\psi_{old}$) is determined by a mask $\mathbf{s}_{\psi_{\text{old}}}$ computed by a scaled and displaced Sigmoid function:

$$s_{\psi_{old}}^j = \text{Sigmoid}(b(\tilde{\alpha}^j \cdot \alpha_{v_{\text{BMU}},\psi_{\text{old}}}^j - 0.5)) \tag{7}$$

where $\tilde{\alpha}$ is the spike variable inferred from $\mathbf{x}$ and $\boldsymbol{\alpha}_{v_{\text{BMU}},\psi_{\text{old}}}$ the spike variable of the best-matching unit $v_{\text{BMU}}$ (the component with the largest posterior) on the snapshot of the past SOM. $b$ scales and sharpens the Sigmoid function towards a gated function (Tonolini et al., 2020). Note that neither $\mathbf{s}_{\psi_{\text{old}}}$ or $v_{\text{BMU}}$ is fixed; they are functions of the unknown spike variables $\tilde{\alpha}$ and $\boldsymbol{\alpha}_{v_{\text{BMU}},\psi_{\text{old}}}$. We now use this to (1) maintain consistency on latent dimensions for shared semantic factors (if any), and (2) prevent entangling new semantic factors into the shared latent dimensions.

**Reusing Shared Semantic Factors:** To teach the model to reuse latent dimensions corresponding to previously-learned semantic factors on new data $\mathbf{x}$, we ask the VAE encoder $q_\phi$ to be consistent with its past snapshot $q_{\phi_{\text{old}}}$ in the shared dimensions when inferring from the new data:

$$\mathcal{L}_{\text{newz}} = \mathcal{L}_c[S(q_\phi(\mathbf{z}|\mathbf{x})) \odot \mathbf{s}_{\psi_{old}}, S(q_{\phi_{\text{old}}}(\mathbf{z}|\mathbf{x})) \odot \mathbf{s}_{\psi_{old}}], \tag{8}$$

where $\odot$ is element-wise multiplication.

**Disentangling New Semantic Factors:** Now since the past model does not necessarily know how to generate new data $\mathbf{x}$, how does it teach the new model? Intuitively, given the relation between $\mathbf{x}$ and the past data as determined in Eqn. (7), even though the past model does not know how to generate the new $\mathbf{x}$, it would know how to generate from those shared semantic dimensions in $\mathbf{x}$. Therefore, as illustrated in Fig. 2(b), if we take a sample $\mathbf{z}_{v_{\text{BMU}},\psi_{\text{old}}}$ from the mixture component $v_{\text{BMU}}$, and replace its latent values with those inferred from $\mathbf{x}$ at the shared latent dimensions, the model should know how to generate from this combined latent vector $\mathbf{z}_{\text{com}}$ in a way consistent to the past generator:

$$\mathbf{z}_{v_{\text{BMU}},\psi_{\text{old}}} = S(p_{\psi_{\text{old}}}(\mathbf{z}|v_{\text{BMU}})), \quad \mathbf{z}_{\text{com}} = \mathbf{z}_{v_{\text{BMU}},\psi_{\text{old}}} \odot (\mathbf{1} - \mathbf{s}_{\psi_{old}}) + S(q_\phi(\mathbf{z}|\mathbf{x})) \odot \mathbf{s}_{\psi_{old}}, \tag{9}$$

$$\mathcal{L}_{\text{newx}} = \mathcal{L}_c[S(p_{\theta_{\text{old}}}(\mathbf{x}|\mathbf{z}_{\text{com}})), S(p_\theta(\mathbf{x}|\mathbf{z}_{\text{com}}))]. \tag{10}$$

Intuitively, if the model entangles new semantic factors into the shared dimensions, it will be penalized as the past decoder does not know how to generate from these new factors. As we will demonstrate, this constraint – uniquely made possible by CUDOS – is critical in continual disentanglement.

**Summary:** In summary, the overall loss for CUDOS is:

$$\mathcal{L} = \mathcal{L}_{\text{ELBO}} + \gamma_1 \mathcal{L}_{\text{old}} + \gamma_2 \mathcal{L}_{\text{newz}} + \gamma_3 \mathcal{L}_{\text{newx}}, \tag{11}$$

where $\gamma_1, \gamma_2$, and $\gamma_3$ are weighting hyperparameters. This extends the foundation objective function to continual learning settings, which promotes the sharing of shared semantic factors between new and past data environments, and the disentangling of new semantic factors.

## 4 EXPERIMENTS AND RESULTS

We evaluated CUDOS on (1) a split version of 3DShapes (Burgess & Kim, 2018) for quantitative evaluation of continual disentanglement, (2) MNIST(LeCun et al., 1998), Fashion-MNIST(Xiao et al., 2017), and their moving versions in (Achille et al., 2018), and (3) split-CelebA (Liu et al., 2015).

**Split-3DShapes:** We quantitatively evaluated the continual disentanglement of past and new representations in a split version of 3Dshapes with two sub-sets: The first one only had red floor and wall, and the second added all floor colors except red. A successful continual learning of representations is expected to continually learn the new factor of floor color while reusing the others learned in the first set. We compared CUDOS to four groups of baselines: (1) naive VAE (Kingma & Welling, 2014), naive TC-VAE (Chen et al., 2018), and VAE with gradually increased capacity (Burgess et al., 2017) without an explicit mechanism to combat catastrophic forgetting, (2) unsupervised continual learning reliant on generative replay (Achille et al., 2018; Ramapuram et al., 2020) and heuristically-defined masks of active latent dimensions (Achille et al., 2018), (3) unsupervised continual learning using a mixture of Gaussian in the latent space (Rao et al., 2019), and (4) a continual learning version of VQ-VAE (Oord et al., 2017) (a prototype-based method similar to SOM) and TC-VAE (Chen et al., 2018) (a representative disentangling VAE) with replay mechanism $\mathcal{L}_{\text{old}}$. Experiments on each model were executed at least 5 times.

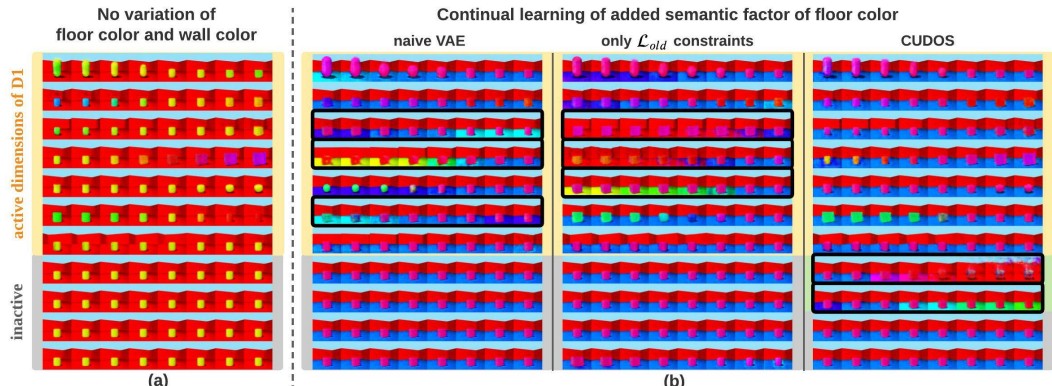

Figure 3: Continually learning a split version of 3DShapes where the variations of floor colors were absent initially and appeared later. Black bounding boxes annotate in which latent dimensions the new semantic factors are learnt. (a): Traversing each latent dimension after training on data with no floor color variations. (b): Traversing each dimension after training on the data with floor color variations, where comparison of CUDOS with baseline methods shows the improved ability to disentangle new semantic factors into previously inactive dimensions.

| Method | MIG ↑ | MIG-sup ↑ | $I(\mathbf{z}_{\text{past}}; \text{factors}_{\text{new}})$ ↓ | Δ reconstruction loss ↓ |
|---|---|---|---|---|
| Naive VAE | 0.229±0.052 | 0.228±0.086 | 1.220±0.295 | 0.351±0.044 |
| Naive TC-VAE | 0.295±0.141 | 0.377±0.169 | 1.113±0.158 | 0.355±0.144 |
| Burgess et al. (2017) | 0.192±0.029 | 0.131±0.018 | 1.154±0.092 | 0.300±0.037 |
| Continual TC-VAE | 0.138±0.112 | 0.254±0.109 | 1.312±0.668 | 0.006±0.004 |
| Continual VQ-VAE | 0.115±0.045 | 0.226±0.045 | 1.470 ±1.236 | 0.008±0.002 |
| Achille et al. (2018) | 0.162±0.057 | 0.234±0.078 | 0.639±0.200 | 0.025±0.010 |
| Rao et al. (2019) | 0.095±0.075 | 0.091±0.037 | 0.886±0.084 | 0.042±0.015 |
| Ramapuram et al. (2020) | 0.197±0.056 | 0.296±0.073 | 1.115±0.222 | **0.003±0.003** |
| CUDOS | **0.242±0.047** | **0.326±0.053** | **0.024±0.029** | 0.021±0.013 |

Table 1: Quantitative metrics for disentanglement. MIG: mutual information gap (Chen et al., 2018). MIG-sup: supplement mutual information gap (Li et al., 2020). $I(\mathbf{z}_{\text{past}}; \text{factors}_{\text{new}})$: mutual information between already active dimensions and the new factors. Δ reconstruction loss: the relative change for reconstructing past data.

Choosing suitable disentanglement metrics is vital as different metrics may measure different aspects of the disentanglement (Locatello et al., 2020; Zaidi et al., 2022). Here we consider two types of metrics. First, to form a complete measurement of the one-to-one relationship between latent dimensions and semantic factors, we chose MIG-sup (Li et al., 2020) that penalizes learning multiple semantic factors into the same dimensions in combination with the complementary MIG (Chen et al., 2018). Second, to focus on the disentanglement of sequentially-arrived semantic factors, we compute the mutual information $I(\mathbf{z}_{\text{past}}; \text{factors}_{\text{new}})$ between active dimensions for past data and new factors in the new data. Ideally, if a model manages to disentangle new semantic factors into dimensions not used by previous data, $I(\mathbf{z}_{\text{past}}; \text{factors}_{\text{new}})$ should be close to 0. Finally, we also compute the relative change of reconstruction loss for past data to measure forgetting.

As shown in Fig.3(a), major semantic factors such as object shape, size, and color were learned from the first subset. When a new data stream is introduced (shown in Fig.3(b)), while all models were able to reuse most of the latent dimensions corresponding to previously-learned semantic factors, all baseline models entangled new semantic factors – the floor color – with these dimensions. In contrast, CUDOS was able to not only reuse shared latent dimensions, but also disentangle new ones into previously inactive dimensions. Note that we include results of Ramapuram et al. (2020) in Fig.3 as an example of the baseline models that have constraints on replayed data, annotated by $\mathcal{L}_{\text{old}}$. Visual traversing results of other comparison models can be found in Appendix.D. Additional results on different sequences of split-3DShapes can be found in Appendix I.

Quantitatively, as summarized in Table. 1, CUDOS witnessed the lowest $I(\mathbf{z}_{\text{past}}; \text{factors}_{\text{new}})$ among all comparison models, as well as the best disentanglement performance as measured by MIG and MIG-

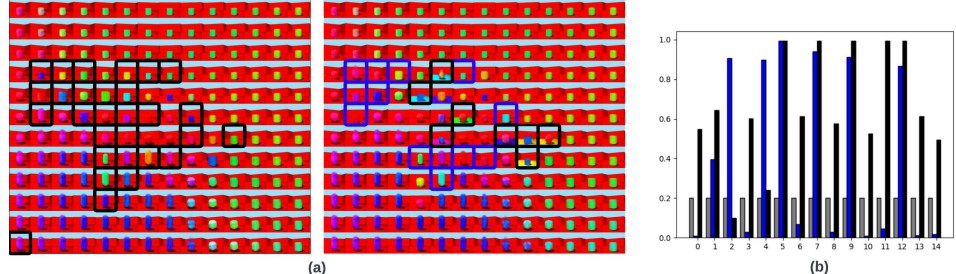

Figure 4: SOM results on split-3DShapes. (a) SOM prototypes learnt after the first (left) and second (right) data environments. Black boxes label prototypes associated with current (black) and replayed data samples (blue), while the rest of the prototypes are mixture-interpolated. (b) Average spike parameter $\alpha_k$. Prototypes of the second data environment (black) shared active dimensions from the first data environment (blue) with added dimensions. Gray represents inactive dimensions.

Figure 5: Continual learning of Moving Fashion-MNIST and MNIST.

Table 2: $R^2$ score on new data & its change $\Delta$ from that on old data prior to continual learing.

| 3DShapes | baselines | CUDOS |
|---|---|---|
| Scale $R^2$ ↑ | 0.05±0.07 | **0.58±0.13** |
| Orientation $R^2$ ↑ | 0.08±0.05 | **0.93±0.02** |
| $\Delta$ Scale ↑ | -0.92±0.10 | **-0.13±0.13** |
| $\Delta$ Orientation ↑ | -0.81±0.15 | **-0.00±0.01** |
| Moving-MNIST | baselines | CUDOS |
| X-pos $R^2$ ↑ | 0.67±0.09 | **0.70±0.06** |
| Y-pos $R^2$ ↑ | **0.75±0.05** | 0.66±0.23 |
| $\Delta$ X-pos ↑ | 0.053±0.13 | **0.054±0.08** |
| $\Delta$ Y-pos ↑ | **0.01±0.06** | -0.15±0.27 |

sup. This carefully designed experiment provided clear evidence that CUDOS is able to continually disentangle new semantic factors without entangling them with shared ones learned from the past. It also showed that this cannot be achieved by naively using generative replay to extend existing disentangling or prototype-based VAE (*e.g.*, TC-VAE and VQ-VAE) into continual versions. More discussion of differences between static and continual disentanglement can be found in Appendix J.

Fig. 4 (a) shows that the SOM-mixture continually updated a summary of the relation between past (blue box) and new data environments (black box) based on their active latent dimensions (b). T-SNE plots of the learned latent representations after continual learning as an alternative way to visualize the relationships among data are provided in Appendix G.

**Moving MNIST & Fashion-MNIST:** We then tested CUDOS on sequences of Fashion-MNIST, MNIST, and moving versions of them similar to that presented in Achille et al. (2018). Fig.5 presented results for one sequence: Moving Fashion-MNIST → Moving MNIST → Fashion-MNIST. As shown, CUDOS was able to disentangle new semantic factors (green boxes), while re-using those learned from the past (red boxes). For instance, the positional semantic factors learned from Moving Fashion-MNIST were reused while learning Moving MNIST.

**CelebA:** We further designed experiments for continually learning over split versions of CelebA data. Fig.6 provide examples of results obtained by two different splits: by age (top), and by bangs (bottom). Traversing results on selected dimensions showed that CUDOS was able to reuse latent dimensions for previous semantic factors while learning new ones related to the new attributes.

**Benefits on downstream tasks:** To evaluate the usefulness of the disentangled representations learnt by CUDOS, we focused on shared tasks related to the continually-learned semantic factors shared among past and new data environments, including predicting the scale and orientation of 3DShapes, and predicting the X- and Y- positions of Moving-MNIST. For each task, we identified the corresponding active latent dimensions after learning the first data environment, and continually trained linear regressors to predict the ground-truth labels using these active latent dimensions: the

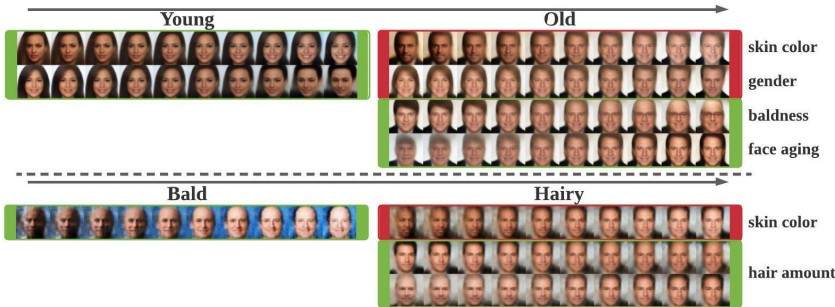

Figure 6: CelebA split by age (top row) and bangs (bottom row). The green boxes highlight the newly learned semantic factors, and the red boxes highlight the reused ones.

|  | moving fashion | moving MNIST | fashion |
|---|---|---|---|
| active | **0.74** | **0.85** | **0.86** |
| inactive | 0.4 | 0.17 | 0.35 |

| Method | MIG ↑ | MIG-sup ↑ | $I(\mathbf{z}_{past}; factors_{new}) \downarrow$ | $\Delta$ reconstruction ↓ |
|---|---|---|---|---|
| VAE+SOM+$\mathcal{L}_{old}$ | 0.128±0.026 | 0.125±0.029 | 0.969±0.238 | 0.011±0.001 |
| VAE+SS+$\mathcal{L}_{old}$ | **0.302±0.121** | 0.293±0.119 | 0.782±0.765 | **0.005±0.103** |
| VAE+SOM+SS+$\mathcal{L}_{old}$ | 0.217±0.072 | 0.240±0.082 | 0.648±0.268 | 0.024±0.004 |
| VAE+SOM+SS+$\mathcal{L}_{old}$+$\mathcal{L}_{newz}$ | 0.216±0.036 | 0.320±0.086 | 0.072±0.050 | 0.024±0.004 |
| CUDOS (above+$\mathcal{L}_{newx}$) | 0.242±0.047 | **0.326±0.053** | **0.024±0.029** | 0.021±0.013 |

Table 4: Top: Continual digit classification accuracy (testing) based on active or inactive dimensions. Bottom: Ablation study. SS: spike-and-slab density. $\mathcal{L}_{old}/\mathcal{L}_{new}$: constraints on replayed/new data.

intuition is that, if new semantic factors are entangled into these shared dimensions, the performance of the regressor will decrease during continual learning. For comparisons, Achille et al. (2018) and Ramapuram et al. (2020) were used as baselines and their results aggregated for 3Dshapes, and Ramapuram et al. (2020) was used as the baseline for Moving-MNIST. Table 2 summarizes the $R^2$ scores of each task obtained on new data along with their changes $\Delta$ from the $R^2$ scores obtained on old data prior to continual learning. As shown, CUDOS achieved best final $R^2$ score for scale and orientation regression, along with minimum drop of performance over the course of continual learning. For Moving-MNIST, CUDOS and baselines achieved similar results on X-Y position regression, with in general less significant performance drop in comparison to 3DShapes. We argue that this is because X-Y position regression on a clean background is an easier task, than scale and orientation regression on a more complex data environment like 3DShapes. Additionally, Table 4 (top) shows digit-classification performance on Moving-MNIST using active and inactive dimensions, which suggested that active semantic factors for the digit data environments are learned properly.

**Ablation study:** Table. 4 (bottom) presents a detailed ablation study on the contribution brought by the different ingredients within CUDOS. We did not include results from VAE + $\mathcal{L}_{old}$ because they are represented by the work of Ramapuram et al. (2020) as reported in Table. 1. As shown, the sparsity introduced by spike-and-slab distribution plays a significant role in improving the disentanglement ability of CUDOS. While SOM alone does not appear to improve the general disentangling ability of the model, it does seem to help disentangle new semantic factors from previously learned ones; more importantly, it is a necessary component for learning the relational structure of data to guide disentanglement, *i.e.*, for enabling $\mathcal{L}_{newx}$ in Equation (10). Indeed, the combined introduction of SOM and $\mathcal{L}_{newx}$ brings significant improvement in the ability of CUDOS to disentangle new semantic factors from previously learned ones, as it is evident in the improvement achieved in MIG-sup and $I(\mathbf{z}_{past}; factors_{new})$. More implementation details can be found in Appendix.C.

**Conclusion:** In this paper, we demonstrated that an overlooked key ingredient to continual unsupervised learning of representations is to exploit the relational structure of data based on their underlying active semantic factors. We presented CUDOS, a novel VAE with self-organizing spike-and-slab mixtures, to address this. A limitation of the present form of CUDOS is that continual disentangling partly depends on the teacher-student strategy, leaving the snapshot timing a hyper-parameter. We will investigate sequential Bayesian inference as a potential solution to this problem as a future work.

ACKNOWLEDGEMENT

This work was supported by National Heart, Lung, and Blood Institute (NHLBI) grant R01HL145590, and National Institute of Nursing Research (NINR) grant R01NR018301.

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

## A   Optimizing $\psi$ from ELBO

To make the partial derivative of $\psi$ clearer, we first rewrote the last two terms of ELBO in Eqn. (3) as:

$$
\begin{aligned}
&- \mathbb{E}_{p_\psi(w|\mathbf{z})}[D_{KL}(q_\phi(\mathbf{z}|\mathbf{x})||p_\psi(\mathbf{z}|w))] - \mathbb{E}_{q_\phi(\mathbf{z}|\mathbf{x})}[D_{KL}(p_\psi(w|\mathbf{z})||p(w))] \\
&= \int_{p_\psi(w|\mathbf{z})} p_\psi(w|\mathbf{z}) \int q_\phi(\mathbf{z}|\mathbf{x}) \log \frac{p_\psi(\mathbf{z}|w)}{q_\phi(\mathbf{z}|\mathbf{x})} d\mathbf{z}dw + \mathbb{E}_{q_\phi(\mathbf{z}|\mathbf{x})} \int p_\psi(w|\mathbf{z}) \log \frac{p(w)}{p_\psi(w|\mathbf{z})} dw \\
&= \int_{q_\phi(\mathbf{z}|\mathbf{x})} q_\phi(\mathbf{z}|\mathbf{x}) \int p_\psi(w|\mathbf{z}) \log p_\psi(\mathbf{z}|w) d\mathbf{z}dw - \int_{p_\psi(w|\mathbf{z})} p_\psi(w|\mathbf{z}) \int q_\phi(\mathbf{z}|\mathbf{x}) \log q_\phi(\mathbf{z}|\mathbf{x}) d\mathbf{z}dw \\
&\qquad\qquad\qquad\qquad\qquad + \mathbb{E}_{q_\phi(\mathbf{z}|\mathbf{x})} \int p_\psi(w|\mathbf{z}) \log \frac{p(w)}{p_\psi(w|\mathbf{z})} dw \\
&= \mathbb{E}_{q_\phi(\mathbf{z}|\mathbf{x})} \int p_\psi(w|\mathbf{z}) \log \frac{p_\psi(\mathbf{z}|w)p(w)}{p_\psi(w|\mathbf{z})} dw - \mathbb{E}_{p_\psi(w|\mathbf{z}),q_\phi(\mathbf{z}|\mathbf{x})}[\log q_\phi(\mathbf{z}|\mathbf{x})] \\
&= \mathbb{E}_{q_\phi(\mathbf{z}|\mathbf{x})}[\log p_\psi(\mathbf{z})] - \mathbb{E}_{p_\psi(w|\mathbf{z}),q_\phi(\mathbf{z}|\mathbf{x})}[\log q_\phi(\mathbf{z}|\mathbf{x})]
\end{aligned}
\tag{12}
$$

Therefore the optimized $\psi^*$ can be solved by:

$$
\psi^* = \mathrm{argmax}_\psi \mathrm{ELBO} = \mathrm{argmax}_\psi \mathbb{E}_{q_\phi(\mathbf{z}|\mathbf{x})}[\log p_\psi(\mathbf{z})]
\tag{13}
$$

## B   Moving MNIST and Moving fashion-MNIST dataset

Based on the original MNIST and fashion-MNIST dataset, we created the moving version of them by: (1) Resize the original 28*28 images to 36*36 images, (2) Place the original 36*36 images image at the top-left of a 64*64 black background. (2) Apply translation for both x and y axis with values [5,10,15,20,25].

## C   Experiments detail

**Hyper-parameters settings and implementation strategy:** We set $\gamma_1 = 0.25, \gamma_2 = 1, \gamma_3 = 0.35$, $b = 10$ in all experiments. Snapshot of the model is updated every $\tau = 1500$ iteration steps. Regarding weights in Eqn 11, in our experiments, generally, we are trying to avoid certain parts of the loss function becoming too large or small, and we found a rule-of-thumb weight value described above across different datasets. The most tricky part is the weighting for the constraints of old data and new data. We found a slightly higher weight on new data can achieve better continual disentanglement results. We reason that the constraint on new factors is more difficult compared with reconstructing old data, and therefore emphasizing more on that (higher weight) can be helpful. In general, we didn't find obvious differences within around 30% percent changes of each weight. Putting them too high (larger than the ELBO term) will affect the original continual learning of new data. Metrics in Table. 1 are calculated in a setting where the boundary of data environments is known.

**Network-architecture:** Table. 5 shows the network architecture for all our experiments, where BN stands for batch normalization, Conv stands for convolution layer, T-Conv stands for transposed convolution layer, FC stands for fully-connected layer, ELU stands for Exponential linear unit activation.

Code is available at `https://github.com/Zhiyuan1991/CUDOS_release`.

## D   Traversing results on split-3DShapes

Here we presented additional traversing results for baseline methods, along with their $I(\mathbf{z}_{\text{past}}; \text{factors}_{\text{new}})$ value.

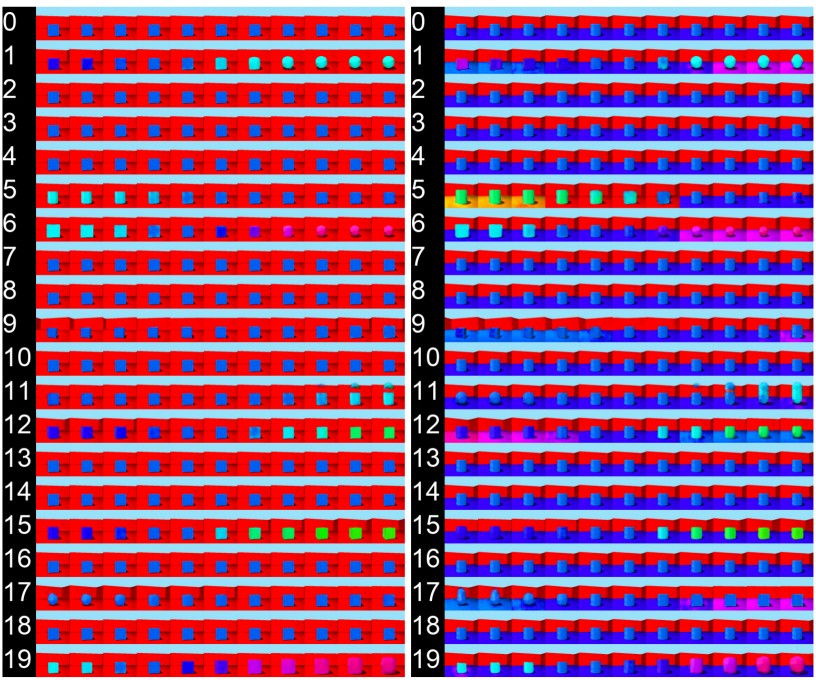

Figure 7: Burgess et al. (2017), $I(\mathbf{z}_{\text{past}}; \text{factors}_{\text{new}})$: 1.185

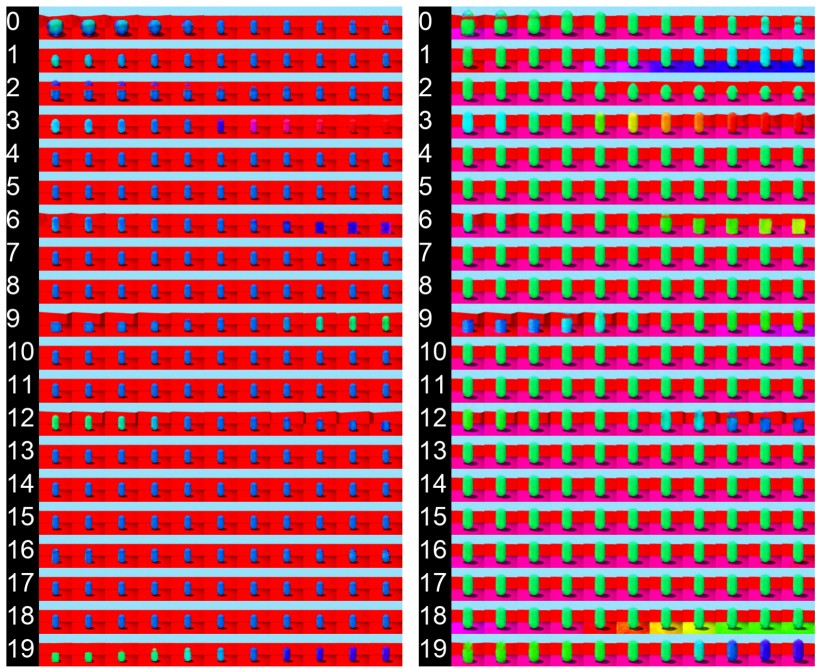

Figure 8: Achille et al. (2018), $I(\mathbf{z}_{\text{past}}; \text{factors}_{\text{new}})$: 0.602.

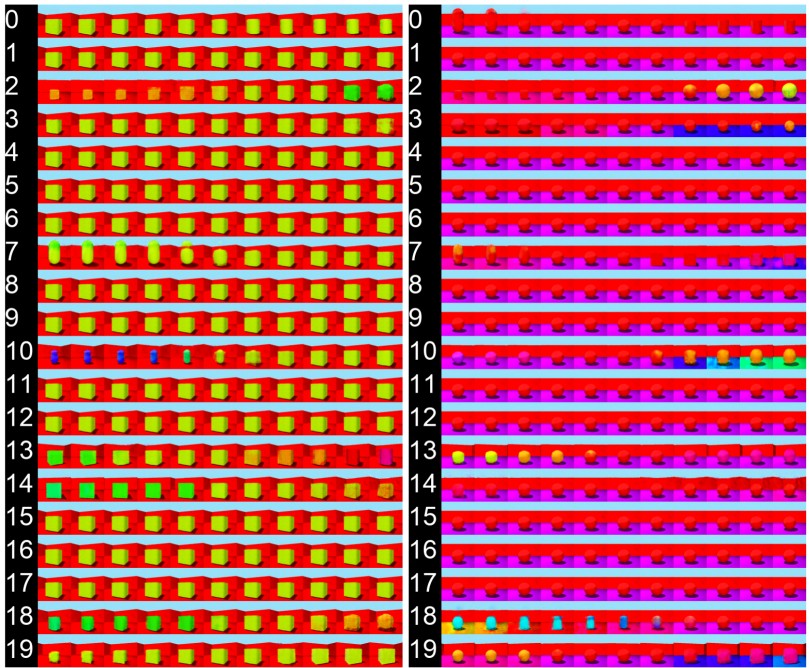

Figure 9: Ramapuram et al. (2020), $I(\mathbf{z}_{\text{past}}; \text{factors}_{\text{new}})$: 0.877

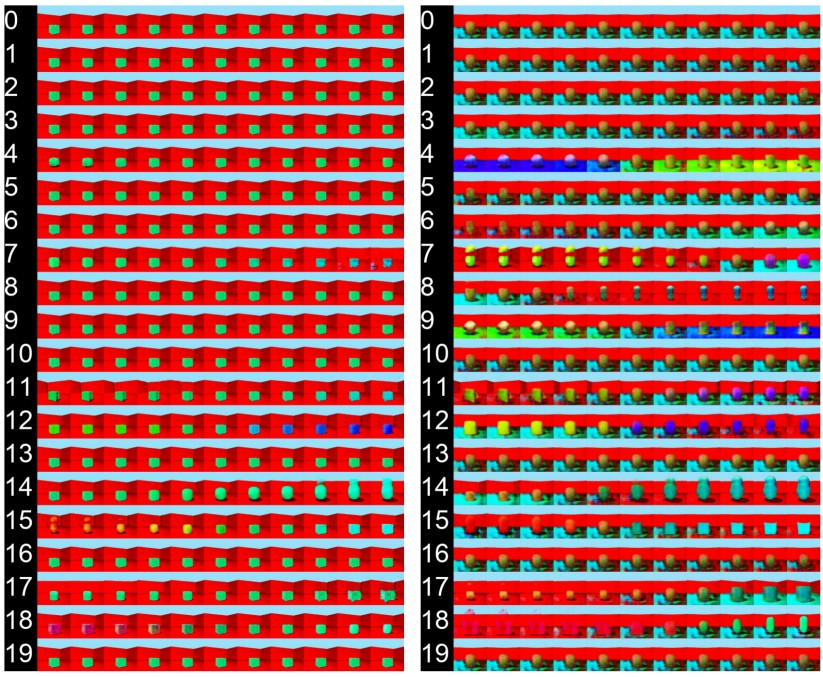

Figure 10: Continual VQ-VAE, $I(\mathbf{z}_{\text{past}}; \text{factors}_{\text{new}})$: 1.402

| Encoder Layer | Encoder parameters | Decoder Layer | Decoder parameters |
|---|---|---|---|
| Conv1 | 32,4,4, strides=2, BN, ELU | FC1 | 1024, BN, ELU |
| Conv2 | 64,4,4, strides=2, BN, ELU | FC2 | 4*4*256, BN, ELU |
| Conv3 | 128,4,4, strides=2, BN, ELU | T-Conv1 | 128,4,4, strides=2, BN, ELU |
| Conv4 | 256,4,4, strides=2, BN, ELU | T-Conv2 | 64,4,4, strides=2, BN, ELU |
| FC1 | 1024, BN, ELU | T-Conv3 | 32,4,4, strides=2, BN, ELU |
| FC2 | 1024, BN, ELU | T-Conv4 | image channels,4,4, strides=2 |
| FC3 | latent dims*2 | | |

Table 5: Network architecture

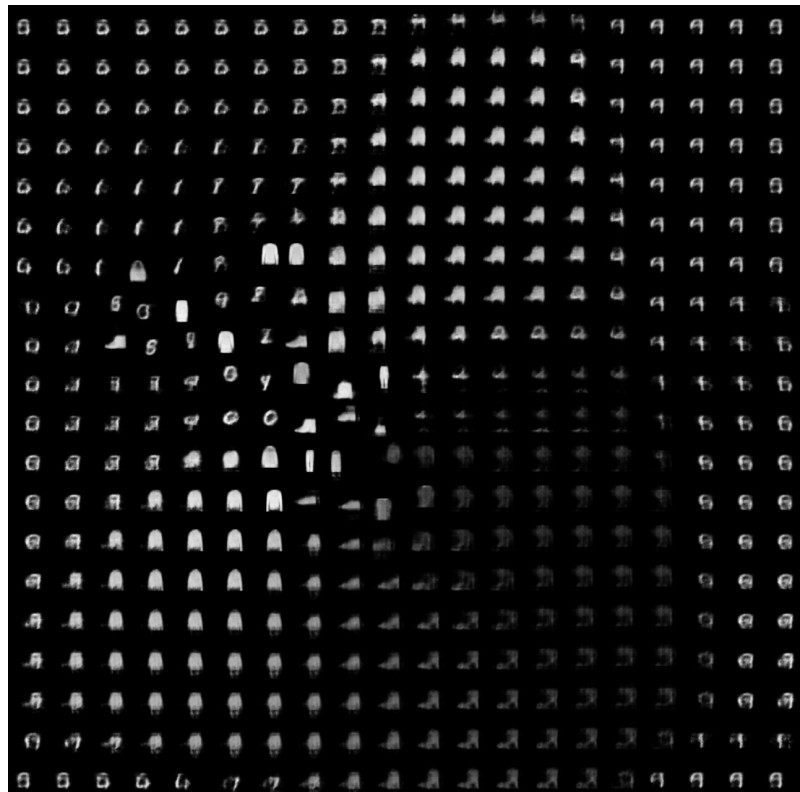

Figure 11: Prototypes learnt for continual learning on Moving-Fanshion-MNIST to Moving-MNIST to MNIST.

## E  SOM PROTOTYPES

Additional SOM prototypes learnt for Moving-MNIST are shown in Fig 11. In the SOM, the presented model was able to remember and accumulate old representations, e.g., fashion digits, while learning new representations, e.g., number digits, Additionally, the shared representations, the X-Y position, were changing smoothly among prototypes.

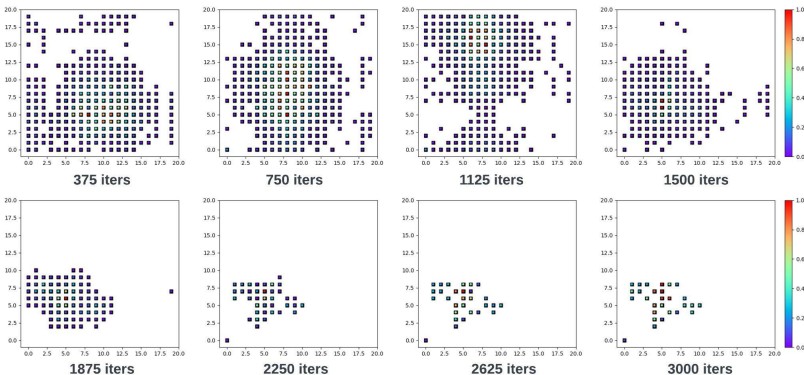

Figure 12: Mapping density on SOM of the first data environment of split-3DShapes during training.

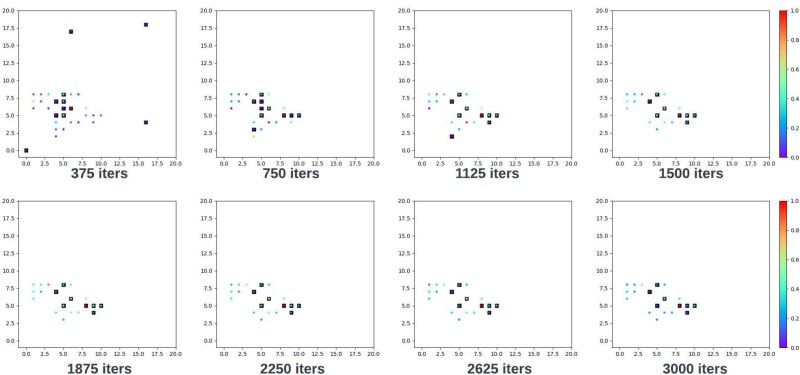

Figure 13: Mapping density on SOM of the second data environment of split-3DShapes during training. Dots without black boundaries are replay data.

## F  DATA ENVIRONMENTS MAPPING ON SOM DURING TRAINING

In Fig 12 and Fig 13 we presented additional data environments' mapping density on SOM during continual training.

## G  T-SNE FOR 3DSHAPES FACTORS

Here we presented additional T-SNE plots for the learned latent representations of our model (colored by the generative factors of 3DShape) after continual learning of split-3DShape as shown in Fig 3. As shown in Fig.14, some factors such as floor color (10 classes) and shape (4 classes) formed good clusters while some more difficult factors such as orientation (15 classes) and scale (8 classes) formed fewer discriminative clusters.

## H  TRAVERSING RESULTS ON MOVING-MNIST

Here we presented additional traversing results for baseline methods on Moving-MNIST.

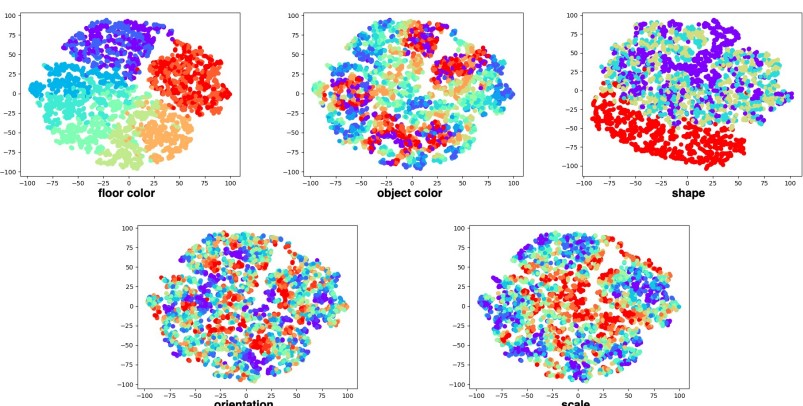

Figure 14: T-SNE plots for 3DShapes factors.

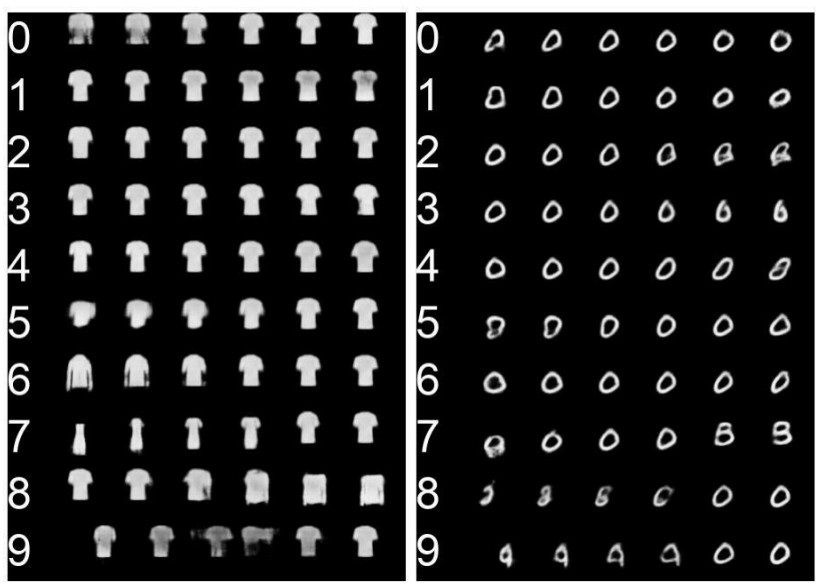

Figure 15: Burgess et al. (2017).

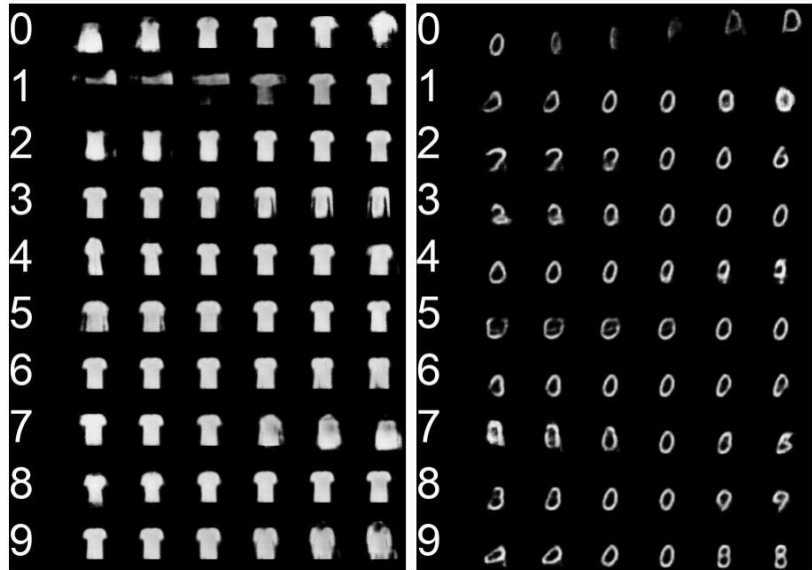

Figure 16: Achille et al. (2018).

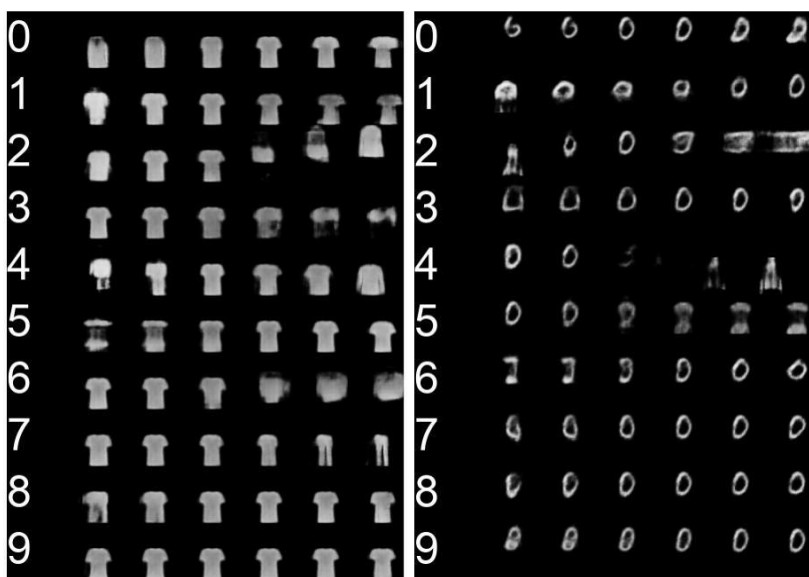

Figure 17: Ramapuram et al. (2020).

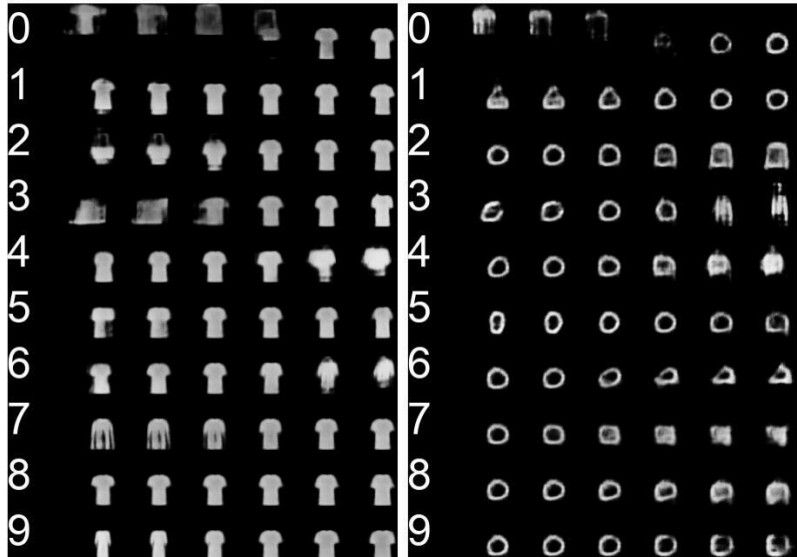

Figure 18: Continual VQ-VAE.

# I    ADDITIONAL EXPERIMENTS OF SPLIT-3DSHAPES

Here we first presented additional experiments of split-3DShapes with two splitting versions that each has three data environments. The first version, as shown in Fig 19, was starting with no floor and wall color variations and then continually added them. The second version, as shown in Fig 20, was starting with no scale and wall color variations and then continually added them. For most latent dimensions during continual learning, CUDOS was able to reuse those corresponding to previously learned semantic factors and disentangle new ones into previously inactive dimensions.

Next, we presented a reversed sequence of split-3DShape, where the first data environment includes all semantic factors and the second one includes a subset that doesn't have the floor color variations (only red remained). As shown in fig 21, our model was able to reuse all previous semantic factors without any expansion of latent space. Furthermore, our model was able to remember how to generate the variations of the missing floor color in the second data environment after continual training.

# J    MORE DISCUSSION OF DIFFERENCES BETWEEN STATIC AND CONTINUAL DISENTANGLEMENT

Continually disentangling sequentially-arrived semantic factors is fundamentally different from disentangling a static dataset where the model sees all semantic factors at once. When semantic factors are presented sequentially, the model needs to make sure that it 1) does not forget previously learned factors, 2) is able to re-use latent dimensions corresponding to shared factors, and 3) prevents new ones to be entangled into the shared ones. None of these challenges are addressed by state-of-the-art (SOTA) disentangling VAEs such as FactorVAE (Kim & Mnih, 2018) and TC-VAE (Chen et al., 2018). The continual setting sees significant challenges such as forgetting (including forgetting learned semantic factors) and, as in other continual learning problems, cannot be expected to see similar disentanglement performance to those reported on static settings. That's why the non-continual baselines in Table 1, including naive VAE, naive TC-VAE, and Burgess et al. (2017), all showed weaker performance than what would have been expected in a static learning setting. Furthermore, because these models are not designed for continual learning, they will also face challenges such as catastrophic forgetting. We can take a closer look at this by taking naive TC-VAE versus continual TC-VAE (with generative replay mechanism) as an example. As shown in Table 1, naive TC-VAE achieved overall better MIG and MIG-sup scores because they can forget about previous data and

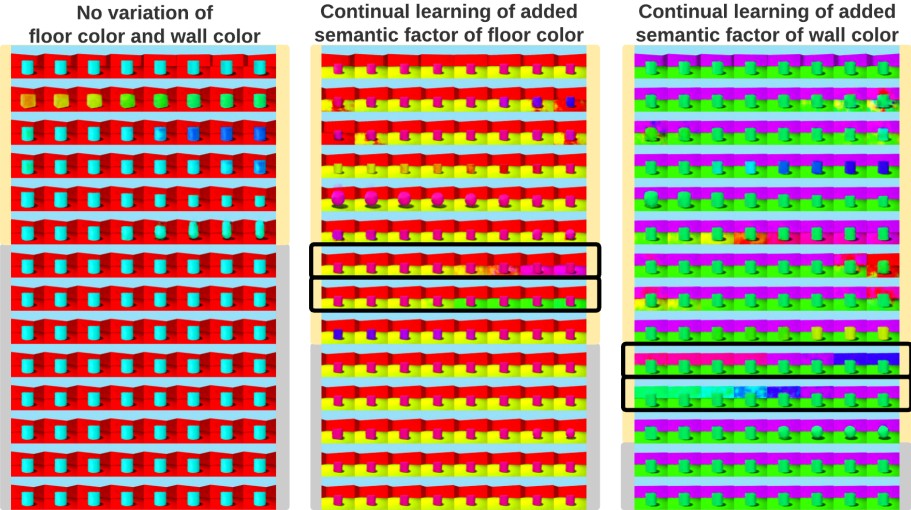

Figure 19: Continually learning a split version of 3DShapes where the variations of floor and wall colors were absent initially and appeared later. Each row of images is traversing each latent dimension after training on data environments titled above. Black bounding boxes annotate where the new semantic factors are learnt.

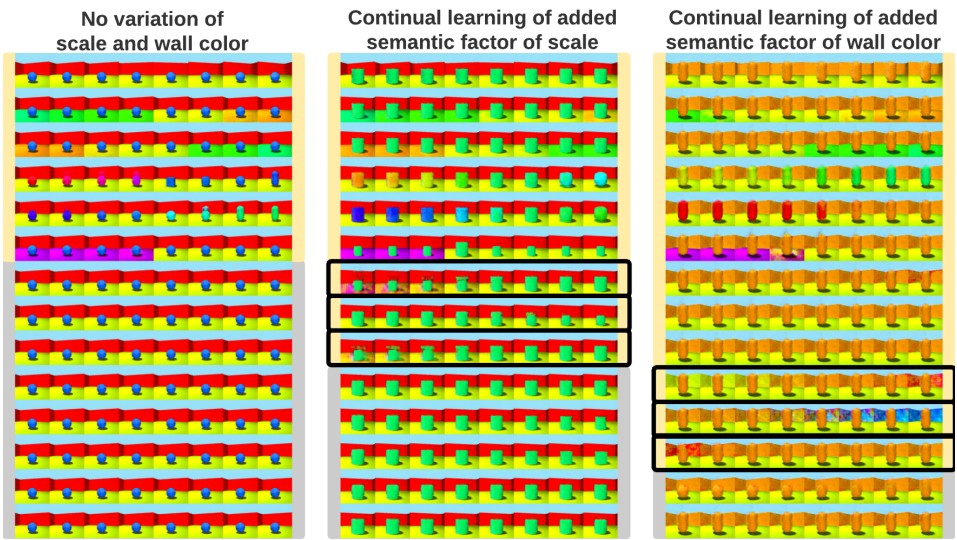

Figure 20: Continually learning a split version of 3DShapes where the variations of scale and wall colors were absent initially and appeared later. Each row of images is traversing each latent dimension after training on data environments titled above. Black bounding boxes annotate where the new semantic factors are learnt.

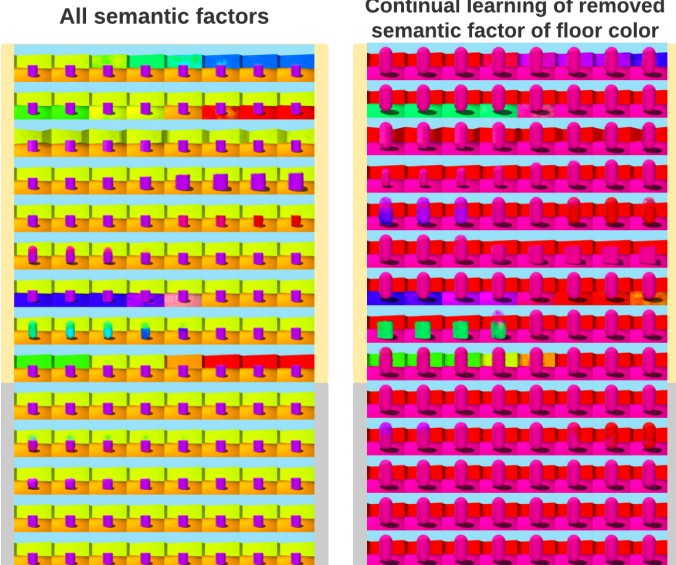

Figure 21: Continually learning a reversed sequence of split-3DShape where all semantic factors were presented in the beginning but later the variation of floor color was removed (only red remained). Each row of images is traversing each latent dimension after training on data environments titled above.

focus on disentangling the new data (in split 3DShapes, the two sequentially-presented datasets share many latent factors but one). Therefore, forgetting previous semantic factors did not create a large performance drop in MIG and MIG-sup scores. This catastrophic forgetting however is reflected in the reconstruction loss, which increased from 2365 to 3207 (around 35% change) on the previous data. In addition, the $I(\mathbf{z}_{\text{past}}; \text{factors}_{\text{new}})$ metric further shows that naive TC-VAE is not able to separate new semantic factors from previously-learned latent dimensions, but rather achieved relatively high disentangling by simply forgetting previous factors and learning on the new data itself. This is similar to naive VAE's performance.

By extending TC-VAE to a continual learning setting with generative replay, the continual TC-VAE was able to remember how to reconstruct the previous data, where the reconstruction loss only increased from 2376 to 2395 (0.7% change). However, its disentanglement performance including MIG and MIG-sup dropped. This demonstrates the aforementioned challenges of disentangling sequentially-arrived semantic factors (while remembering the previous factors at the same time) that are fundamentally different from disentangling all seen factors at once. It also shows that such challenges cannot be addressed by either naive SOTA disentangling VAEs, or simply extending these SOTA disentangling VAEs into a continual setting (via standard techniques such as generative replay). We believe these provide further evidence for the contribution of the presented work.

