# OpenReview forum: "Continual Unsupervised Disentangling of Self-Organizing Representations"
_ICLR.cc/2023/Conference — ICLR 2023 notable top 25%_

### Official Review · Reviewer_ARpk · 2022-10-20

**Confidence:** 3
**Correctness:** 3
**Technical Novelty And Significance:** 3
**Empirical Novelty And Significance:** 3
**Recommendation:** 6

**Clarity, Quality, Novelty And Reproducibility:**

The originality of the work is relatively good, there are many interesting innovative components of the proposed methodology.

The quality is not satisfying enough, as there are concerning issues with experiments (see above).

The clarity could be further improved. Many details about model training are lacking.


**Strength And Weaknesses:**

Strength
- It is interesting to utilize sample-2-sample relationship for disentanglement, and it is interesting to see how it can be applied in the continual learning regime, where semantic factors can possibly be expanded.
- The paper presented solid mathematical/theoretical contributions by formulating the variational inference process with Bayesian-SOM and spike-and-slab distribution. The presented method seems to have several unique advantages because of the selected priors.

Weaknesses

1. Major concern: experiments and methods for benchmarking aren't sufficient for demonstrating the strength of the method.
- The proposed method seems to have relatively weak disentanglement performance when compared with previously existing methods. Specifically, [1] reported several models’ MIG scores on 3Dshapes (the non-splitting version), where FactorVAE has an average of 0.3, AnnealedVAE has an average of 0.4. They also reported relatively high model variance, where a naive VAE can achieve a MIG score of as high as 0.8. It is understandable that the paper reported relatively low variance as each model is run 5 times and there might exist different settings for randomness. However, it is not convincing that the model actually disentangles well, as it does not outperform naive VAE by a large margin, and some other VAE methods that are designed for disentanglement study can easily outperform the proposed method.
- The benchmarking models are selected with certain bias. Only model2 (beta-VAE by Burgess) and model4 (life-long disentanglement by Achille) are specifically designed for disentanglement (correct me if I am wrong). All other methods are continual learning methods. It is understandable to include many continual learning methods, as the proposed method is also based on continual learning techniques. However, as the method is motivated to address the disentanglement problem, it should consider more methods (e.g. VAEs), and especially state-of-the-art methods, that are specifically designed for disentanglement (e.g. FactorVAE, TC-VAE, DIP-VAE) to have a fair comparison.
- Although the evaluation of disentanglement has been a long-standing problem [1, 2], using the mutual information I(z_{past}; factors_{new}) seems to be a particularly biased one as the proposed method is optimized on similar components.

2. Related work related to disentanglement learning is lacking
- There are many different variants of VAE that are designed for disentanglement that are not considered and mentioned. The original beta-VAE paper [3] isn’t cited. [4-6] are VAEs that are mentioned above. [7-9] are more recent unsupervised methods that consider sparse coding or separatable latent factors.
- When doing unsupervised disentanglement learning, it is also recommended to clearly demonstrate the assumptions [1].

3. Motivation isn’t strong enough
- It isn’t totally convincing to me why one would like to perform disentanglement under the continual learning setting. I think it would be beneficial if the authors could stress less about how it is the first work that evaluates the disentanglement scores for continually learned latents, but stress why it would be important instead.

4. Minor comments:
- The details of the Split-3DShapes dataset aren’t clear. How many data points are within each subset? Why it is separated in this particular way?
- How are the hyper-parameters selected? I did not find any details about the model selection procedure. What are the training/validation/testing sets? (also for the benchmark models)
- I am not sure if the important details for reproducing the models are included. What is the model learning rate?
- Does the network architecture apply to all benchmark models?


[1] Locatello, Francesco, et al. "Challenging common assumptions in the unsupervised learning of disentangled representations." international conference on machine learning. PMLR, 2019.

[2] Mathieu, Emile, et al. "Disentangling disentanglement in variational autoencoders." International Conference on Machine Learning. PMLR, 2019.

[3] Higgins, Irina, et al. "beta-vae: Learning basic visual concepts with a constrained variational framework." (2016).

[4] Kim, Hyunjik, and Andriy Mnih. "Disentangling by factorising." International Conference on Machine Learning. PMLR, 2018.

[5] Chen, Ricky TQ, et al. "Isolating sources of disentanglement in variational autoencoders." Advances in neural information processing systems 31 (2018).

[6] Kumar, Abhishek, Prasanna Sattigeri, and Avinash Balakrishnan. "Variational inference of disentangled latent concepts from unlabeled observations." arXiv preprint arXiv:1711.00848 (2017).

[7] Rhodes, Travers, and Daniel Lee. "Local Disentanglement in Variational Auto-Encoders Using Jacobian $ L_1 $ Regularization." Advances in Neural Information Processing Systems 34 (2021): 22708-22719.

[8] Liu, Ran, et al. "Drop, swap, and generate: A self-supervised approach for generating neural activity." Advances in Neural Information Processing Systems 34 (2021): 10587-10599.

[9] ​​Horan, Daniella, Eitan Richardson, and Yair Weiss. "When Is Unsupervised Disentanglement Possible?." Advances in Neural Information Processing Systems 34 (2021): 5150-5161.


**Summary Of The Paper:**

This paper attempts to perform unsupervised disentanglement with continual representation learning methods. Specifically, they designed a variational autoencoder (VAE) that exploits the relationships between datapoints based on their latent factors. The model learns the latent representations of data by applying a self-organizing mixture of spike-and-slab priors (which is topologically informed) to the VAE. The proposed method is validated on many disentanglement datasets (3DShapes, variants of MNIST, and celebA), where the proposed method showed superior disentanglement performance when compared to other continual learning algorithms.


**Summary Of The Review:**

I suggest rejection for this paper. Although there are many interesting and innovative components of the proposed approach, the evaluation of the proposed method isn’t convincing as many benchmark models that should be considered are not considered. The introduction (motivation) and the related work section could also use major editions to improve clarity. I am willing to increase the score for this paper if sufficient revision is made accordingly to address my major concerns.

---

> ### Author Response · Authors · 2022-11-17
> **Response to reviewer ARpk part-1**
>
> Dear reviewer, thank you for your time and suggestion.
>
> In response to the main criticism, we must clarify that the goal of the paper is NOT to use continual learning as a technique to improve disentanglement in a standard static setting, but to address disentangling **sequentially-arrived semantic factors** continually.  This is fundamentally different from disentangling a static dataset where the model sees all semantic factors at once: when semantic factors are presented sequentially, the model needs to make sure that it 1) does not forget previously learned factors, 2) is able to re-use latent dimensions corresponding to shared factors, and 3) prevent new ones to be entangled into the shared ones. None of these challenges are addressed by SOTA disentangling VAEs such as FactorVAE and TC-VAE.
>
> **Disentanglement performance**
>
> The “relatively weak disentanglement performance” pointed out by the reviewer is exactly because the results reported are from a “continual learning” setting, e.g., where the model is sequentially presented with the split version of 3DShapes and sees different semantic factors sequentially. The performance range (0.3-0.8 MIG) noted by the reviewer, in comparison, is obtained on disentangling a static full version of 3DShapes, where the model sees all semantic factors at once. The continual setting sees significant challenges such as forgetting (including forgetting learned semantic factors) and, as in other continual learning problems, cannot be expected to see similar performance to those reported on static datasets. That’s why the non-continual baselines in Table 1, including naive VAE and Burgess et al, all showed weaker performance than what would have been expected in a static learning setting. We have added an additional baseline of vanilla TC-VAE (see below) and its performance in continual setting further provides evidence for this.
>
> We further would like to point out that, while CUDOS improved over naive VAE moderately in MIG, it significantly improved MIG-sup, I(z_past,factor_new), and reconstruction loss. For reconstruction loss, it shows that naive VAE models will forget how to reconstruct previous data in a continual learning setting. In terms of disentanglement, while MIG measures how much a semantic factor is encoded into different latent dimensions, MIG-sub and I(z_past, factor_new) measures how much the same latent dimension encodes different semantic factors. These measure two complementary aspects of disentanglement and, while dimensional independence encouraged by TC-VAE emphasizes the former aspect, this paper emphasizes the latter. The significant improvements in these metrics demonstrates the intended contribution of CUDOS.

---

> > ### Author Response · Authors · 2022-11-17
> > **Response to reviewer ARpk part-2**
> >
> > **Disentanglement baselines**
> >
> > For the same reason, we originally did not include SOTA disentangling VAEs such as FactorVAE and TC-VAE as baselines because they are not designed for continua learning. In comparison, VAE-based continual representation learning works such as Rao2019 and Ramapuram2020 are most related to our work. In addition, although these papers did not focus on continual disentanglement, they have discussed the importance of disentanglement. That’s why we chose them as the main baselines in this paper.
> >
> > However, as suggested by the reviewer, we did add comparison with SOTA disentangling VAEs (specifically TC-VAE given the limited time). Because these models are not designed for continual learning, they face challenges such as catastrophic forgetting as demonstrated in the following experiments. Therefore, in addition to the naive TC-VAE (directly apply TC-VAE on continual data stream), we also included a continual extension of TC-VAE (we modified the framework to add generative replay mechanism to solve catastrophic forgetting as we did to continual VQ-VAE). We ran the same experiments of splitted-3DShape datasets on naive TC-VAE and continual TC-VAE. The results on the split 3DShapes following the same experimental setting as in our paper are as follows:
> >
> > |			      |    MIG↑    |        MIG-sup↑      |      I(z_past;factors_new)↓    |   ∆reconstruction loss↓ |
> > |---------------------|------------|---------------------|---------------------------------|----------------------------|
> > |Naive TC-VAE       |        0.295±0.141  |         0.377±0.169     |      1.113±0.158         |             0.355±0.144
> > |Continual TC-VAE |  0.138±0.112       |     0.254±0.109     |      1.312±0.668              |        0.006±0.004
> > |CUDOS                 |        0.242±0.047 |           0.326±0.053      |     0.024±0.029           |           0.021±0.013
> >
> > Here, naive TC-VAE achieved overall better MIG and MIG-sup scores because they can forget about previous data and focus on disentangling the new data – in split 3DShapes, the two sequentially-presented datasets share many latent factors but one; therefore, forgetting previous semantic factors did not create a large performance drop in MIG and MIG-sup scores. This catastrophic forgetting however is reflected in the reconstruction loss, which increased from 2365 to 3207 (around 35% change) on the previous data. In addition, the I(z_past, factor_new) metrics further shows that naive TC-VAE is not able to separate new semantic factors from previously-learned latent dimensions (but rather achieved relatively high disentangling by simply forgetting previous factors and learning on the new data itself). This is similar to naive VAE’s performance.
> >
> > By extending TC-VAE to a continual learning setting with generative replay, the continual TC-VAE was able to remember how to reconstruct the previous data, where the reconstruction loss only increased from 2376 to 2395 (0.7% change). However, its disentanglement performance including MIG and MIG-sup dropped. This demonstrates the aforementioned challenges of disentangling sequentially-arrived semantic factors (while remembering the previous factors at the same time) that is fundamentally different from disentangling all seen factors at once. It also shows that such challenges cannot be addressed by either naive SOTA disentangling VAEs, or simply extending these SOTA disentangling VAEs into a continual setting (via standard techniques such as generative replay). These results are now added to Table 1 of the revised manuscript. We believe these provide further evidence for the contribution of the presented work, and we thank the reviewer for pushing us to consider these comparisons.

---

> > > ### Author Response · Authors · 2022-11-17
> > > **Response to reviewer ARpk part-3**
> > >
> > > **Rationale of  I(z_past;factors_new) as a metric**
> > >
> > > As mentioned by the reviewer, the evaluation of disentanglement has indeed been a long-standing problem and, as far as we know, all existing disentanglement metrics are each focused on a specific perspective of disentanglement, mostly motivated by the particular disentangling method proposed. For instance, TC-VAE emphasizes the independence of latent dimensions; accordingly its proposed MIG metric penalizes two latent dimensions having similar mutual information to a given semantic factor (i.e. violating the independence constraint). In continual disentanglement, we are particularly interested in how much new semantic factors are entangled in the previously-used dimensions. Existing metrics do not capture this, which is how the presented metric of  I(z_past;factors_new) is motivated. However, it is important to clarify that neither this metric nor any mutual information were directly used in the loss for optimizing our model, so we do not believe this metric is particularly biased. We hope our discussion alleviates the concerns of the selection of metrics, as in page 7 starting with “Choosing suitable disentanglement metrics is vital as…”.
> > >
> > > **Related works and motivation**
> > >
> > > We added citations to representative disentangling VAEs in paragraph 2, page 1 in the Introduction. As mentioned above, comparisons and discussion of vanilla disentangling VAE (and its continual version with generative replay) are also added to the Experiment section of the revised manuscript.
> > >
> > > Disentangled representation learning, as a long-standing research topic, has demonstrated various benefits in generative modeling and downstream tasks. As increasing recent works investigate unsupervised representation learning in a continual learning setting, we believe it to be important to investigate the challenges and solutions to achieve disentanglement in such settings. We have included this motivation in the Introduction, and demonstrated its potential to benefit downstream tasks in Table 2 (similar to those in achille2018).
> > >
> > > **Others**:
> > >
> > > For splitted-3DShape, there is 3840 data in the first split (only had red floor and wall) and 34560 data in the second split (added all floor colors except red). There is no particular reason for doing this and other versions of splitting with different factors can be tried. We have added additional experiment results with different splitting versions in Appendix I.
> > >
> > > Hyper-parameters are selected by manual grid-search. All models shared the same encoder/decoder architecture as described in appendix. All images are resize to 64*64 to fit this architecture. Learning rate for VAE is 5e-4 shared for all experiments. Code is sent to reviewers now.

---

> ### Comment · Reviewer_ARpk · 2022-11-17
> **Increase score**
>
> Thank you for providing detailed explanation and clarification in response to my comments. With the additional clarification and experiments, I think it is fair to increase my original score from 3 to 6. I would suggest the authors include the results and discussion in response part-2 (the naive TC-VAE results and the comments re differences between naive TC-VAE and continual TC-VAE) in the paper (Appendix or main text) to further motivate the method.

---

> > ### Author Response · Authors · 2022-11-18
> > **Response to reviewer ARpk**
> >
> > Dear reviewer, we are glad we have solved most of your concerns and thank you for re-evaluating our works with an increased score. As you suggested, we have added the discussion of the difference between static and continual disentanglement in appendix J, and we found this is indeed a valuable discussion to further enrich this paper's content.  We also added naive TC-VAE’s results to the main text (Table 1) and referred the reader to the discussion in appendix J from the main text. Thank you for your time and efforts again.

---

### Official Review · Reviewer_z1ce · 2022-10-24

**Confidence:** 3
**Correctness:** 3
**Technical Novelty And Significance:** 3
**Empirical Novelty And Significance:** 3
**Recommendation:** 8

**Clarity, Quality, Novelty And Reproducibility:**

### Quality
The submission is of high quality. The problem of continual unsupervised disentangled learning is well-defined and well-motivated. The architecture is derived in a principled manner and justified by an ablation study. The evaluation takes into account the most relevant competing methods. As mentioned before, the manuscript would benefit from another round of editing.

### Novelty
While the individual components of the proposed architecture are well established, the particular configuration is novel to the best of my knowledge. Shifting the focus towards unsupervised learning is a valuable contribution to the field of continual learning, dominated by supervised discriminative learning.

### Reproducibility
The appendix contains details on generating the datasets, a blueprint of the network architecture, and values of the relevant hyperparameters. This information should be enough to reproduce the results. The authors promise to release the code, but the URL in the appendix is a placeholder.

### Clarity
- The paper is well structured and formatted, but grammatical mistakes, typos, and inconsistent punctuation make it occasionally hard to follow (see minor points).
- What is meant by "environment" in the second paragraph?
- "The latent variable z from the k-th mixture component" <--- Is this correct wording?

Minor:
- The keyword "disentanglment" should be corrected
- The discussion on "generative-replay" in the first paragraph seems unrelated to the rest of it.
- "Expanding learned semantic factors, in part, *results* naturally from"
- "While the common strategy of generative replay teaches a model what semantic factors to use on the replayed data" <--- I'm not sure if the previous paragraphs clearly explain this.
- "learning of progress in continual unsupervised representation learning is relatively limited" sounds strange.
- Using the notation $p(z|w_k=1)$ in eq.1 might be better for consistency.
- Shouldn't the first equation in eq.2 be $q(z,w|x) = q_\phi(z|x)p_\psi(w|z)$?
- "We follow the theory in (Gepperth & Pf¨ulb, 2021)" <--- This can be explained with a few words
- Including more explanation in Fig2 caption would be better
- Typos
  - determie ---> determine
  - "In existing works, reusing semantic factors **are** mainly attempted by a teacher-student like approach (...)"
  - "To overcome this, we argue that the model needs to learn **two critical knowledge** (...)"
  - "toplogoically" (page 3),
  - "stick-and-slab" (page 4)
  - "CUODS" (page 8)
  - "we presents" (page 16)


**Strength And Weaknesses:**

### Strengths
- It is a thoughtful and careful approach to designing the model architecture. Authors start with a well-defined problem, identify a problem in existing literature (the independence assumption) and propose a creative solution. The use of specific components is well-motivated, and the intuitions are verified through ablation studies. The loss formulation is principled and well-argued.
- The paper is very well-written. The sentences are crystal clear and the ideas are communicated in a great level of detail. The transitions are also very smooth. I really enjoyed reading the work!

### Weaknesses
- The final loss term (11) is rather complicated, involving many moving parts. Although each term is individually motivated, ablation studies displaying their impact are needed.
- Qualitative evaluation can be improved. The authors provide quantitative evidence that their method performs continual disentanglement, but it would be informative to dig deeper into the results shown in Figure 3. In particular, introducing novel factors (not limited to floor colour) in sequence would shed more light on the learning dynamics and make a more convincing argument that the model discovers, reuses, and expands latent semantic factors. Another extension (arguably out of the scope of the submission) would be to use the model as a generative replay module in existing continual learning methods and check if the disentangled representation provides benefits for downstream tasks like classification or semantic segmentation.


### Questions
- I don't immediately see why sparse coding enables the discovery of "active semantic factors" in an auto-encoder. In particular, why are these "semantic factors" instead of, e.g., "active features"? This could be elaborated more.
- I'm not sure how exactly $\alpha$'s are inferred for streaming data. Precisely, why do we maintain a set? Here, does each "semantic factor" correspond to one latent dimension? How does this connect to Figure 1?
- Why do we need to address the fact that "generative replay lacks mechanisms to constrain newly arrived data"?
- Does the method maintains a fixed number of SOM snapshots? How are they updated as learning proceeds?
- If eq.8 is optimized perfectly, then would we still need to optimize eq.9? It seems to me that both objectives serve the same purpose?

**Summary Of The Paper:**

This paper aims to achieve continual, disentangled, and unsupervised representation learning where the representations can be reused and expanded upon newly arriving data. For this, the authors propose to model the relational structure of continuously arriving data based on their "active semantic factors", which allows for the discovery of new semantic factors while recycling (the old) shared ones. The two pillars of the methodology are (i) self-organizing maps (SOMs) to maintain a relational data structure and (ii) sparse spike variables to model the underlying active semantic factors in each mixture component. The method is shown to achieve outstanding continual disentanglement of latent semantic factors.

**Summary Of The Review:**

The paper presents a novel, well-motivated method that tackles an important and under-explored aspect of continual learning. While the qualitative evaluation could be more thorough, the submission in the current form would already be a valid contribution to the continual learning and representation learning community. I recommend acceptance.

---

> ### Author Response · Authors · 2022-11-17
> **Response to reviewer z1ce**
>
> Dear reviewer, we appreciate your time and many insightful comments.
>
> **Additional ablation**
>
> To investigate the influence of each module of our models, in the original submission, we provided an ablation study in table 4 (page 9) that focused on the impact of the 1) SOM, 2) spike-and-slab density (SS), and 3) L_new that includes constraints on the new data (L_old was in all versions because it is a basic component necessary to prevent catastrophic forgetting). In response to the reviewer’s comment, we added additional ablation studies to separate the impact of L_newx and L_newz. Results support the benefits of both terms, and are now added to Table 4 of the revised manuscript.
>
> **Additional qualitative evaluation**
>
> As suggested, we conducted additional qualitative experiments in appendix I. We run additional experiments of split-3DShapes with two splitting versions that each has three data environments. The first version, as shown in Fig.19, is starting with no floor and wall color variations and then continually adding them. The second version, as shown in Fig.20, was starting with no scale and wall color variations and then continually added them. For most of latent dimensions during continual learning, CUDOS was able to reuse those corresponding to previously learned semantic factors, and disentangle new ones into previously inactive dimensions.
>
> **Response to additional questions**
>
> - We thank the reviewer for pointing this out. Indeed, the sparse encoding enables the discovery of “active latent dimensions” which is then related to semantic factors in the data. For continual disentanglement, we would like to re-use the same latent dimensions for semantic factors shared among data environments and, if newly arrived data have semantic factors not used in previous data, we would like them to be encoded to previously unused latent dimensions. We have revised the manuscript throughout to make this clear.
>
> - We originally tested a formulation where alpha is inferred per data sample by q(alpha|x). In experimentation, however, we discovered that there did not seem to be sufficient statistical strength for the model to infer active latent dimensions for each data sample. Because it is natural to assume that active semantic factors (and thus the underlying active latent dimensions) are shared by data from the same data environments, we chose to leverage the statistical strength of a set of data that shares latent semantic factors (thus shared active latent dimensions) to help the inference of alpha.
>
> - Without constraining newly arrived data, we discovered that the model can use the previously active latent dimensions for different semantic factors. This is because it receives no training signal on how to re-use or avoid previously-used active dimensions. We have added clarifications to this sentence in 3.2.3. Of the revised manuscript.
>
> - Regarding the question on SOM snapshot, we only maintain one snapshot of SOM during training. This is the same for the snapshot of encoder and decoder.
>
> - The optimization of Eqn.8 and Eqn.9 is a great question. In theory, if Eqn.8 is optimized perfectly, and if the decoder works the same as before for the existing active latent dimensions, we believe that it will ask the model to not entangle new semantic factors to these reused dimensions. In practice, however, we observed that with Eqn.8 alone the entanglement of new factors into existing active dimensions still occur. We reason that this is because Eqn.8 constrains only the encoder and there is no training signal on the decoder for how it should behave on the new data. Because encoder and decoder are optimized together in the VAE framework, we then added Eqn. 9 to constrain the behavior of the decoder on the new data.  From our experimental experience and as shown in the newly-added ablation study, optimizing Eqn.9 along with Eqn.8 will be helpful for continual disentanglement.
>
> - We use “environment” to refer to data with different distributions, following recent literature. We have also corrected minor comments raised. Unfortunately, we didn’t find a way to fix the keyword “ disentanglement” right now in OpenReview. We will fix it as soon as possible once we find out how to edit that.

---

### Official Review · Reviewer_ZXCc · 2022-10-25

**Confidence:** 4
**Clarity, Quality, Novelty And Reproducibility:** I am satisfied with the clarity, qual…
**Correctness:** 4
**Technical Novelty And Significance:** 3
**Empirical Novelty And Significance:** 3
**Recommendation:** 6

**Strength And Weaknesses:**

Strength:

- The CUDOS can treat continually-arrived data independently, without knowing how they are related based on the underlying semantic factors.

- This paper has conducted a large number of experiments to testify to the disentangling ability of CUDOS.

- This work mitigates the catastrophic forgetting from a novel perspective of generative replays from SOM-mixture.


Weakness:

- More visualization like TSNE can further clarify the disentangling ability of CUDOS.



**Summary Of The Paper:**

In this paper, the author shows that an overlooked key ingredient to continual unsupervised learning of representations is to exploit the relational structure of data based on their underlying active semantic factors. This paper proposes a novel VAE with self-organizing spike-and-slab mixtures called CUDOS.

**Summary Of The Review:**

This paper presents CUDOS, a novel VAE with self-organizing spike-and-slab mixtures.

---

> ### Author Response · Authors · 2022-11-17
> **Response to reviewer ZXCc**
>
> Dear reviewer, we appreciate your time and efforts.
>
> To visualize the disentangling ability of CUDOS, SOM serves a similar purpose toT-SNE as a way to visualize the clustering of the latent embedding. Examples of this SOM-based visualization were provided in Fig 4 (at pape 8) and 11 (in appendix E, page 15) for 3DShape and Fashion MNIST and MNIST.
>
> Additionally, as suggested by the reviewer, in appendix G we added T-SNE plots for the learned latent representations of our model (colored by the generative factors of 3DShape) after continual learning of split-3DShape as shown in Fig.3. Some factors such as floor color (10 classes) and shape (4 classes) formed good clusters while some more difficult factors such as orientation (15 classes) and scale (8 classes) formed less discriminative clusters.

---

### Official Review · Reviewer_NqT4 · 2022-10-29

**Confidence:** 4
**Correctness:** 3
**Technical Novelty And Significance:** 3
**Empirical Novelty And Significance:** 3
**Recommendation:** 6

**Clarity, Quality, Novelty And Reproducibility:**

The paper makes some novel contributions in continual disentanglement. Readability of the paper could be improved. Code is not available to reproduce the results.

**Strength And Weaknesses:**

Strengths:
------------
+ **Motivation:** Each of the issues related to continual disentanglement: catastrophic forgetting, expanding, and reusing latent dimensions are dealt with separately using three objective functions that are added to the traditional ELBO objective in a Variational Auto Encoder. Such isolation gives more control over the latent space to achieve better disentanglement.
+ **Methodology:** When training on new data environment, CUDOS expands the latent dimension by using new semantic factors that are not entangled with existing active semantic factors (Equation 7). Such expansion is a good addition to the CUDOS methodology to improve the disentanglement score in continual learning.
+ **Experiments:** Experimental results are promising in confirming to the theoretical claims of the paper.

Weaknesses:
----------------
- **Readability:** The paper is challenging to understand in some places. Especially, Sub-Section 3.2 could be more detailed. For example, does the line ‘Generative replay lacks mechanisms to constrain newly arrived data.’ mean CUDOS wants to generate a replay buffer using old semantic factors? Also, some preliminaries on existing work, such as how SOMs are used in VAEs, would make the paper more self-contained.
- **Methodology:** Why is it essential to expand latent dimensions while training in new environments? Why cannot existing/already active semantic factors capture new variations in the data? For example, in Figure 3.b and page 7 of the results section, while I agree that CUDOS uses previous inactive dimensions for learning new semantic factors, naive VAE also generate variations w.r.t. one semantic factor while keeping other semantic factors fixed. That is, naive VAE and CUDOS both produce the desired behavior of disentangling semantic factors. What advantage does CUDOS bring by using previously inactive dimensions?
- **Methodology:** Are semantic factors continually expanded when training in a new data environment? What if active semantic factors of an environment describe a new data environment? Is it guaranteed that CUDOS will not learn/expand the existing semantic factors? How does it affect the training procedure? If it is assumed that a new data environment leads to expanding latent space, the authors should clarify it in the paper and its effect on the training procedure.
- **Experiments:** In the experiments, the reconstruction loss of CUDOS is not the best compared to baselines. Since good generation is an essential aspect of any generative model, this could be a negative aspect of CUDOS. Is this because of multiple objective functions in Equation 11? What is the effect of weighting hyperparameters of Equation 11 on the final output?
- A typo: in the results of Moving MNIST & Fashion-MNIST, it should be Fig. 5 instead of Fig. 4.

**Summary Of The Paper:**

This paper proposes a method for disentangling semantic factors (a.k.a. generative factors) in a continual setting where the observed data from different environments come in a stream. This paper studies continual disentanglement in unsupervised learning, an under-explored area. This paper proposes principled solutions to reusing, expanding, and continually disentangling latent dimensions for each data environment coming in a continual fashion. The overall method is called the Continual Unsupervised Disentangling of self-Organizing representations (CUDOS). CUDOS learns the relationships among active latent dimensions for each data environment and reuses appropriate latent dimensions while training on new data environments. Latent space is modeled as a mixture of self-organizing maps (SOM) where each mixture component is modeled as a spike-and-slab distribution to encourage sparse representation in latent space. This paper proposes separate objective functions for each desideratum of continual disentanglement. Empirical studies show that CUDOS performs better than existing methods on various datasets w.r.t. multiple metrics.

**Summary Of The Review:**

This paper addresses an important and under-explored idea of unsupervised continual disentanglement of the data coming from different data environments. As discussed in the weaknesses, it is unclear what real-world advantage CUDOS brings by expanding latent dimensions (which is a major contribution of CUDOS) for each data environment. I feel experimental results are sufficient to verify the theoretical claims of the paper. Considering the novel contributions, I provide my rating as ‘marginally above the acceptance threshold’.

---

> ### Author Response · Authors · 2022-11-17
> **Response to reviewer NqT4**
>
> Dear reviewer, thank you for your time and suggestions.
>
> **Readability and preliminaries***
>
> Regarding the statement “Generative replay lacks mechanisms to constrain newly arrived data”: Without constraining newly arrived data, we discovered that the model can use the previously active latent dimensions for different semantic factors, causing entanglement. This is because it receives no training signal on how to re-use or avoid previously-used active dimensions. We have added clarifications to this sentence in 3.2.3. of the revised manuscript. We have made some revisions throughout the manuscript according to reviewers’ comments to further improve readability.
>
> Since CUDOS utilizes a Bayesian formulation of SOMs with spike-and-slab densities – it’s essentially a mixture model and its relation with traditional SOM thus has become weak; we thus omitted preliminaries of previous uses of SOMs in VAE due to page limit, but dedicated the space to describe the mixture of spike-and-slab SOM in the presented method instead. In general, as we briefly introduced in related works, SOM has been mainly used as a parallel constraint for learning deep embeddings.
>
> **Necessity to expand latent dimension**
>
> If existing active dimensions capture new variations in the data, semantic factors corresponding to the new variations in the data will be entangled with previously learned semantic factors (encoded in existing active dimensions). As a result, the model will NOT be able to “generate variations w.r.t. one semantic factor while keeping other semantic factors fixed – this is what was demonstrated in Fig. 3b: naive VAEs generate the new variations of “floor colors” (rows 3, 4, 6) together with previously learned semantic factors such as object scale (row 3), orientation (row 4), and object color (row 6). It is not able to generate variations w.r.t. only the new semantic factors while keeping the previously-learned ones fixed, because the new factor is entangled with the previous ones. This is the main challenge our paper is poised to address.
>
> On the other hand, if the semantic factors of a new environment have already been described by previous active dimensions, there is no need to expand and our model is encouraged to reuse these active dimensions. In other words, we didn’t assume that a new data environment must lead to expanding latent space. In experimentation, however, reusing (i.e, not learning/expanding) the existing semantic factors is not guaranteed, and we can occasionally observe that a new dimension is activated for learning the existing semantic factors of new environments, which is an undesired result. Most of the time, our model is able to reuse existing semantic factors and its corresponding active dimensions correctly. To demonstrate this, we have added additional experiments in appendix I with a reversed sequence of split-3DShape, where the first data environment includes all semantic factors and the second one includes a subset that doesn't have the floor color variations (only red remained). As shown in fig.21, our model was able to reuse all previous semantic factors without any expansion of latent space. Furthermore, our model was able to remember how to generate the variations of the missing floor color in the second data environment after continual training. In the future, we want to further enhance the definition and the optimization of the shared dimensions between different environments, which is vital for improving the performance of reusing existing (shared) semantic factors.
>
> ** Reconstruction loss and weighting parameters**
> Indeed, improving continual disentanglement involves some levels of trade-off with reconstruction quality. This is similar to what has been reported in SOTA disentangling literature,  where the improvement of disentanglement performance needs to be traded with reconstruction performance. Similarly, here, while CUDOS is not the best at reconstructing (but the gap is small), it brings significant improvement in its ability to disentangle sequentially presented semantic factors. In our experiments, generally we are trying to avoid certain parts of the loss function becoming too large or small, and we found a rule-of-thumb weight value described in appendix C across different datasets. The most tricky part is the weighting for the constraints of old data and new data. We found a slightly higher weight on new data can achieve better continual disentanglement results. We reason that the constraint on new factors is more difficult compared with reconstructing old data, and therefore emphasizing more on that (higher weight) can be helpful. In general, we didn’t find obvious differences within around 30% percent changes of each weight. Putting them too high (larger than the ELBO term) will affect the original continual learning of new data. We have added these descriptions in appendix C.

---

### Author Response · Authors · 2022-11-17
**Overall Response**

We would like to thank all the reviewers for their encouraging and constructive feedback. While we will be responding to each reviewer in detail, we outline some major additions to the revised manuscript below:

- We have added a new baseline representing SOTA disentangling VAE, i.e., TC-VAE, both in its vanilla and continual-learning version. Experiments further demonstrate that the challenge of disentangling sequentially-arrived semantic factors cannot be addressed by using existing techniques such as generative replay to simply extend SOTA disentangling VAEs to a continual version.
- We have clarified the importance of needing constraints on teaching the encoder and decoder how to behave on new data, and added additional ablations to show the benefits of these two constraints separately.
- We have added several qualitative experiments on different splits of 3DShapes on three sequences of splits (Appendix I), as well as experiments to demonstrate that the model will not expand if the new sequence of data does not introduce new semantic factors (Appendix I).
- We added tSNE visualizations of the latent representations (Appendix G) to corroborate what was shown in the SOM visualization (Fig 4).

Additionally, we have gone through the paper to further improve clarity in response to the reviewers’ comments and questions.

---

### Decision · Program_Chairs · 2023-01-20

**Decision:**

Accept: notable-top-25%

**Justification For Why Not Higher Score:**

The reasons are due to the weaknesses listed in the summary such as readability. The authors are encouraged to make the necessary changes to the best of their ability in the final camera-ready version of the paper.

**Justification For Why Not Lower Score:**

All reviewers are positive on the paper. The strengths I also agree include the well motivated method, the novel way of mitigating the catastrophic forgetting by relative replaying from SOM-mixture, the promising results confirming the theoretical claims. Some reviewers had concerns on readability, missing details, justification of expanding latent dimension, and ablation study.
After rebuttal, the authors addressed most of the concerns along with a revision.
Since the tackled problem is an under-explored area, and the proposed method makes a valid contribution to both continual learning and representation learning communities, I recommend accept.

**Metareview: Summary, Strengths And Weaknesses:**

This paper proposed a method for continual unsupervised learning of disentangled representation, called CUDOS. Unlike previous work that assumes independence between new and past data, CUDOS models the relationship of data in the latent space using a Bayesian formulation of self-organizing maps (SOM),
 encouraging sparse representation and continual disentangling of new semantic factors. Empirical studies show that CUDOS performs better than existing methods on various datasets w.r.t. multiple metrics.

The strengths include the well motivated method, the novel way of mitigating the catastrophic forgetting by relative replaying from SOM-mixture, the promising results confirming the theoretical claims.

The main weakness and concerns are readability, missing details, justification of expanding latent dimension, and ablation study.


**Note From Pc:**

if the above contains the word "oral" or "spotlight" please see: "oral" presentation means -> notable-top-5% and "spotlight" means -> notable-top-25%. As stated in our emails, we are disassociating presentation type from AC recommendations